# Microbial Biosensors for Rapid Determination of Biochemical Oxygen Demand: Approaches, Tendencies and Development Prospects

**DOI:** 10.3390/bios12100842

**Published:** 2022-10-08

**Authors:** Vyacheslav A. Arlyapov, Yulia V. Plekhanova, Olga A. Kamanina, Hideaki Nakamura, Anatoly N. Reshetilov

**Affiliations:** 1Laboratory of Biologically Active Compounds and Biocomposites, Federal State Budgetary Educational Establishment of Higher Education “Tula State University”, 300012 Tula, Russia; 2Pushchino Center for Biological Research of the Russian Academy of Sciences, G.K. Skryabin Institute of Biochemistry and Physiology of Microorganisms, Russian Academy of Sciences, 142290 Pushchino, Russia; 3Department of Liberal Arts, Tokyo University of Technology, 1404-1 Katakura, Hachioji, Tokyo 192-0982, Japan

**Keywords:** microbial fuel cell, biosensors, biochemical oxygen demand (BOD), organic and inorganic polymers, mediators

## Abstract

One of the main indices of the quality of water is the biochemical oxygen demand (BOD). A little over 40 years have passed since the practical application of the first microbial sensor for the determination of BOD, presented by the Japanese professor Isao Karube. This time span has brought new knowledge to and practical developments in the use of a wide range of microbial cells based on BOD biosensors. At present, this field of biotechnology is becoming an independent discipline. The traditional BOD analysis (BOD_5_) has not changed over many years; it takes no less than 5 days to carry out. Microbial biosensors can be used as an alternative technique for assessing the BOD attract attention because they can reduce hundredfold the time required to measure it. The review examines the experience of the creation and practical application of BOD biosensors accumulated by the international community. Special attention is paid to the use of multiple cell immobilization methods, signal registration techniques, mediators and cell consortia contained in the bioreceptor. We consider the use of nanomaterials in the modification of analytical devices developed for BOD evaluation and discuss the prospects of developing new practically important biosensor models.

## 1. Introduction

During the oxidation of organic substrates in an oxygen-containing aqueous medium, the oxygen is consumed by aerobic microorganisms. This is the main effect used in determining the biochemical oxygen demand (BOD) index. The BOD value is most often expressed in milligrams of oxygen consumed per litre of sample during 5 days of incubation at 20 °C (BOD_5_). Also in use is the BOD_ultimate_ index, which indicates the amount of oxygen required for the complete oxidation of organic compounds to carbon dioxide and water as a result of microbial growth, death, decomposition and cannibalism prior to the initiation of the nitrification processes [1]. The BOD_5_ index was chosen as the definitive test of organic pollution of rivers well back in 1908 [2].

The processes of biochemical oxidation of organic substances are coupled with those of microbial vital activity. In a general form, they can be expressed by the following equation:(1)$$C_{x}H_{y}O_{z}N+\left(x+\frac{y}{4}-\frac{z}{2}-\frac{3}{4}\right)O_{2}\to xCO_{2}+\left(\frac{y-3}{2}\right)H_{2}O+NH_{3},$$
(2)$$C_{x}H_{y}O_{z}N+NH_{3}+O_{2}\to C_{5}H_{8}O_{2}N+\left(x-5\right)CO_{2},$$
where *C_x_H_y_O_z_N* is the total ratio of the main elements present in polluted waters, *C_5_H_8_O_2_N* is the average ratio of the main elements in the cellular substance of bacteria. Equation (1) describes the processes during the respiration of microorganisms; Equation (2) delineates the processes of synthesis of cellular substances in microorganisms. The total amount of oxygen consumed for these two processes corresponds to the value of the BOD_ultimate_. The nature of oxygen demand is distorted due to nitrification processes occurring under the action of nitrifying bacteria (*Nitrozomonas*, *Nitrobacter*, etc.). These oxidize nitrogen-containing compounds that may be present in polluted natural and waste waters (of, e.g., food enterprises), and convert nitrogen from the ammonium form into the nitrate form. Nitrification begins approximately on the 7th day of incubation. As the amount of oxygen consumed for nitrification is several times higher than that required for the biochemical oxidation of organic compounds, various inhibitors of the vital activity of nitrifying bacteria are added to the sample during the determination of BOD_10_, BOD_20_, etc. [3,4]. The level of BOD is also influenced by seasonal changes in temperature, which affects the solubility of oxygen and the speed of biochemical processes in microorganisms. Reservoirs with a BOD_5_ of more than 5 mg O_2_/dm^3^ are usually considered dirty [5]. The value of the BOD indicates the degree of wastewaters’ hazard for reservoirs, the ability of surface waters for self-purification and the overall ecological situation in the aquatic environment.

The classical method of BOD assessment is based on the measurement of the consumption of oxygen by microorganisms (Figure 1A) [3,4,6]. The value of BOD is determined by the difference in the dissolved oxygen content in the test water, enriched with dissolved oxygen and infected with aerobic microorganisms, before and after incubation under standard conditions. The method is quite simple and reliable but has a number of drawbacks. The measured BOD concentrations range from 0.5 to 1000 mg O_2_/dm^3^; additional dilutions are carried out for the analysis of samples with high BOD. The measurements can be performed by a specially trained chemical analyst, and the considerable duration of the analysis does not satisfy the modern requirements for analytical methods, since it does not allow real-time monitoring of the degree of wastewater treatment. Besides, the results of the classical method may have a significant spread of obtained data (greater than 20%) due to the use of various microbial populations typical of particular laboratories. These shortcomings have led to the development of a number of new techniques to replace the outdated approach known as the dilution method. The manometric method, associated with pressure difference measurements (since oxygen is bound by microorganisms during the oxidation of various compounds, and the pressure above the water in a closed vessel will decrease), has become widespread in industrial enterprises. This is because it is the easiest to use and makes it possible to determine the BOD in samples without diluting the sample (the range of determined BOD values, 0–700 mg O_2_/dm^3^ without diluting the sample, as compared with 0–6 mg O_2_/dm^3^ for the standard method) [7]. However, the problem of the need for a 5-day incubation of the sample is not solved by this method, either.

For the rapid BOD analysis, investigators develop methods based on the use of biosensors [8,9]. The principle of biosensor analysis is that, interacting with the analyzed substance, a biological material immobilized on a physical/chemical transducer generates an analytical signal that depends on the content of the substance being determined. This signal is registered by the transducer, is processed and then presented as numerical data (Figure 1B).

The first biosensor for the determination of the BOD was developed in 1977 by Isao Karube based on the bacteria *Clostridium butyricum* IF0 3847 [10]. The incubation of the sample and the measurement took about 30 min. In 1979, this group developed a flow system of the BOD biosensor based on the yeast *Trichosporon cutaneum* [11]. Thus, Dr Karube became the founder of a new direction of biosensorics. Since then, many scientists from around the world have contributed to the study of this field and have developed numerous models of biosensors and biofuel cells for BOD determination. Figure 2 shows an increase in the number of publications on this topic in the ScienceDirect database over the past 20 years, which indicates a growing interest in this area of research.

Some of the proposed laboratory models have found practical application and have been commercialized. The purpose of this review is to consider the basic principles of wastewater monitoring using various types of microbial biosensors and microbial fuel cells (MFCs). The main issues that arise in the development of BOD sensors are considered: principles of signal generation, types of biomaterial, methods of its immobilization, approaches to sensor calibration, difficulties that can be encountered when developing such devices, as well as directions and prospects for further research.

## 2. Microorganisms of BOD Biosensors

BOD biosensors most often make use of the following types of receptor elements: pure cultures of microorganisms, mixtures of various pure cultures, or mixtures of mixed cultures such as microbial consortia formed in industrial wastewater sedimentation tanks or activated sludge [12]. From the point of view of the correctness of BOD determination, the use of microbial associations, especially activated sludge, is preferable, since it enables the expansion of the spectrum of organic substances oxidized by the bioreceptor element (Figure 3, [13]). However, association-based biosensors feature poor reproducibility of results over time, which is due to the ongoing change in the composition of associations. The use of single strains of microorganisms can reduce the accuracy of BOD determination due to a narrower spectrum of oxidizable substances; however, it allows for greater stability of measurements [7,14]. Some works propose to solve this problem by using associations of two or three strains of microorganisms and more [13,15,16,17,18]. However, in this case, the problem of maintaining the unchanged composition of associations may persist.

The most commonly used microorganisms for the formation of BOD biosensors include yeast strains *T. cutaneum*, *Arxula adeninivorans*, *Saccharomyces cerevisiae*, *Debaryomyces hansenii*, *Hansenula anomala*, *Torulopsis candida* and bacterial strains *Pseudomonas putida*, *Escherichia coli*, *Bacillus subtilis*, *B. polymyxa*, *Serratia marcescens* [19,20,21,22,23,24]. As a rule, yeasts are capable of oxidizing a wider range of substrates and are more resistant to adverse factors of the environment (temperature, salinity, etc.) [21,25]; at the same time, bacteria make it possible to create more sensitive analyzers [26,27].

Recently, the use of biofilms has been a topical trend in the formation of BOD biosensors’ receptor elements [28,29,30]. Bacterial biofilms are microbial aggregates located in extracellular polymer matrices and attached to the surface of the biosensor pickup. Immobilization of microorganisms on the electrode to form a biofilm is a unique way to solve the problem of expanding the substrate specificity and increasing the stability of the microbial community. The biggest problem of using biofilms was to achieve a stable and efficient electron transfer between microorganisms and the electrode surface. In the last decade, it has been noticed that a new type of microbial architecture, the electroactive biofilm (EAB), spontaneously develops on polarized electrodes. EAB conducts electrons over large distances and performs quasi-reversible electron transfer on conventional electrode surfaces; besides that, it has self-healing properties. EABs have aroused considerable research interest for their use as the basis of biosensors to monitor the environment [29].

Extracellular electron transfer in EABs is closely related to microbial metabolism and energy production. Based on the phylogenetic analysis of electroactive microorganisms used in biosensors and MFCs, it has been concluded that *Shewanella oneidensis* and *Geobacter sulfurreducens* are two model microorganisms which form EABs [31].

Thus, a new type of BOD sensor using a biofilm with *Geobacter* sp. bacteria (Figure 4) has been presented [32]. The sole source of carbon for film growth is ethanol, which makes it possible to grow a biofilm with the this particular microorganism being predominant. At the same time, the presence of other cultures in the biofilm makes it possible to expand the scope of the device and determine the BOD of synthetic wastewaters and dairy industry wastewaters with a reproducibility of 94% and an error of 7.4%. Electroactive biofilms selected in bioelectrochemical systems on an ethanol medium are good candidates for biosensor applications, since they can bind directly to the electrode due to the presence of *Geobacter* sp. Their relatively high diversity also enables their rapid adaptation to different sources of carbon.

## 3. Types of Transducers

### 3.1. Electrochemical Systems for Rapid Analysis of BOD

#### 3.1.1. Biosensor Analyzers of BOD Based on the Registration of Oxygen by Clark Electrode

Most of the BOD biosensors described to date are microbial biosensors based on immobilized whole cells. Their principle of operation is based on the measurement of oxygen uptake by microorganisms immobilized on the surface of the transducer. In 1977, Karube et al. published a paper where they first described a microbial sensor for the determination of the BOD [10]. A feature of the work of such biosensors is that part of the oxygen contained in the layer of immobilized microorganisms is consumed during the oxidation of organic compounds that occurs in the sample. The remaining oxygen penetrates through the gas-permeable membrane and is reduced at the cathode of the oxygen electrode. The current strength in the system is directly proportional to the amount of oxygen reduced at the electrode. After an equilibrium is established between the diffusion of oxygen into the layer of immobilized microorganisms and the rate of endogenous respiration of immobilized microorganisms, the equilibrium (background) current is registered. When a sample of the analyzed water is injected into the measuring cell, organic substances from the sample are utilized by immobilized microorganisms, as the result of which the rate of respiration of cells increases and leads to an increase in the rate of oxygen uptake. In this case, less oxygen penetrates through the electrode membrane to be reduced. The current will decrease until a new equilibrium is established. When a buffer solution for flushing the biosensor is fed into the measuring cell, the rate of endogenous respiration of microorganisms and the initial equilibrium of oxygen flows in the system are restored. The biosensor signal will be proportional to the concentration of readily oxidized organic substances in the sample. A diagram of the design of such biosensors is presented in Figure 5.

Another approach to the design of BOD rapid analyzers are systems based on bioreactors (see the scheme in Figure 6). Such a bioreactor has two holes for the influent and effluent and for the reference electrode. Also, there are the anode and cathode. The reactor is inoculated with sludge from a wastewater treatment plant to form a bioanode, and a nutrient medium for microbial growth is supplied. Application of a certain voltage across the electrodes enables tracing of the changes in the electrochemical characteristics of the anode at the inflow of various carbon sources (including the investigated effluents), which serves as a substrate for microbial biochemical reactions.

BOD biosensors based on bioreactor technology are mainly developed for online monitoring [35,36]. The principle of operation of these biosensors is based on the analysis of oxygen consumption by microorganisms continuously in contact with the analyzed sample (the process diagram is shown in Figure 7, [37]). Wastewater saturated with dissolved oxygen is supplied to the bioreactor. The oxygen electrode measures changes in the intensity of microbial respiration as the result of contact with the sample. Unidentified microorganisms or activated sludge are most often used as biomaterial in such systems [38]. The design of such a biosensor provides a high cell density in the receptor element, which is higher than in natural or waste water due to the continuous supply of organic matter. On the other hand, the microbial community inside the bioreactor is fully adapted to the analyzed organic substrates, which significantly reduces the analysis time [39]. The response time of such analyzers is usually less than 45 min, making it possible to monitor wastewaters in real time.

An advantage of bioreactor-type BOD analyzers is the easy replacement of the transducer without disturbing the activity of biomaterial. Besides, a bioreactor BOD biosensor, as compared with a biosensor based on immobilized biomaterial, usually has more stable characteristics [36]. A disadvantage of such devices is that they are stationary and are not suitable for field measurements. Thus, reactor-type biosensor systems have a strictly defined purpose—continuous monitoring of wastewater treatment processes at the relevant enterprises.

#### 3.1.2. Mediator-Type Biosensor Systems

BOD assays using oxygen sensors are influenced by the content of dissolved oxygen in the sample. The use of mediator biosensors based on the use of compounds capable of reversible oxidation-reduction makes it possible to obtain results independent of the oxygen level in the analyzed sample and, in addition, to create devices quite small in size and weight (Figure 8, [40,41,42]). R&D resources in the field of mediator rapid BOD analyzers are directed in two directions: directly creating BOD biosensors based on the use of mediator electron transfer [43,44,45,46] and modifying the standard method of BOD analysis [47,48,49]. The possibility of modifying the standard method of BOD analysis using mediators was first presented by Pasco and co-authors in 2000 [50]. This approach is based on anaerobic incubation of the analyzed sample in the presence of potassium hexacyanoferrate (III) mediator. The BOD is determined by the amount of potassium hexacyanoferrate (II) formed after incubation.

The development of BOD biosensors using electron transport mediators is one of the most promising areas of research [21,51]. The equilibrium state of the current in such systems is established within several seconds, which ensures a high speed of analysis. Due to the large active surface area of the measuring electrode (which can be further increased by adding nanomaterials, etc.), on which more biological material can be immobilized, significant currents are generated in the mediator microbial sensors that can exceed the currents of the oxygen electrode [51].

A necessary characteristic of biosensors is the possibility of their miniaturization. The use of screen-printing technology to fabricate electrodes makes it possible to produce cheap, disposable sensors based on microbial cells to be used by a wide range of consumers [9,52]. An important advantage of mediator BOD biosensors is the possibility of their use under anaerobic conditions. This is due to the fact that the enzymes of the respiratory chain are regenerated under the action of mediators but not oxygen. A diagram of the structure of such devices is presented in Figure 9.

For operating with eukaryotic microorganisms, it has been proposed to use two-mediator systems. For example, Nakamura et al. [21] created a biosensor based on the microorganisms *S. cerevisiae* and a two-mediator system of potassium hexacyanoferrate (III)–2-methyl-1,4-naphthoquinone (menadione). Menadione can penetrate into the cell and take electrons from NADH and the electron transport chain [53]. Potassium hexacyanoferrate (III), widely used in biosensors [54], was used as the second component of the system. Thus, the created two-mediator biosensor made it possible to determine the BOD_5_ within the range of 6.6–220 mg O_2_/dm^3^. A two-mediator system was used in a biosensor based on *S. cerevisiae*, which made it possible to increase its useful analytical signal [55]. The authors also suggest that lipophilic menadione penetrates into the cells of microorganisms, before then transferring the accepted electrons to potassium hexacyanoferrate (III), which is rapidly oxidized on the electrode (Figure 10). In further studies, a similar system was used to create a biosensor for the analysis of general toxicity [56].

Potassium hexacyanoferrate (III) has a sufficiently large redox potential and, accordingly, serves as a good electron acceptor. At the same time, due to the fact that its molecule is charged, it cannot penetrate the plasma membrane. Thus, potassium hexacyanoferrate (III) interacts mainly with membrane enzymes of microorganisms. On the other hand, the problem with lipophilic mediators like menadione is that they easily penetrate cell membranes, but their potential is usually too low for them to be strong electron acceptors. It should be noted that this approach, based on the creation of two mediator systems, makes it possible to increase the efficiency of electron transfer from cells to the electrode. However, when creating such systems, both the rate of interaction of the mediator with the electrode and the rate of interaction of the mediator with microorganisms should be taken into account [24].

In the recent decade, emphasis on the creation of mediator biosensors has shifted to the field of biosensor design based on redox-active and conductive polymers. Redox-active polymers are compounds in which electron transfer occurs due to the process of redox reactions between the neighbouring fragments of the polymer chain. A diagram of such a process is presented in Figure 11.

Redox-active polymers provide for reagent-free analysis, stability of the sensor and the absence of mediator washout in the course of operation. Promising materials for the creation of electrically conductive gels are non-toxic substances with biocompatibility, such as proteins [57,58] or polysaccharides [59]. Compliance with these requirements is required for immobilization of biomaterial using hydrogels. Polyaromatic compounds, which have semiconductor properties, are often used as the basis for conductive polymers [60,61].

The electrically conductive properties of many polymers are increased by the addition of nanomaterials, e.g., carbon nanotubes [62,63], graphene [64], etc. It should be noted that the use of various nanomaterials to improve the characteristics of amperometric biosensors is a certain trend, which is also used in the creation of mediator BOD biosensors [30,51,57]. Thus, nanomaterial-modified polymers can be used as an immobilizing biocompatible matrix for biological material, which can simultaneously perform the functions of an electron transport conductor.

### 3.2. Optical Biosensor Systems

One of the directions of BOD biosensors is associated with the development of optical detection principles [15,65,66]. As a rule, optical BOD biosensors are based on two main approaches: the use of an oxygen optical sensor based on the measurement of fluorescence quenching, or the use of fluorescent microorganisms in a receptor element. In the former case, microorganisms are immobilized together with oxygen-sensitive fluorescent dyes (Figure 12a); in the latter, the intensity of fluorescence of microorganisms in oxidation of organic substances is measured directly (Figure 12b) [65,67].

In [68], Yoshida and co-authors used three pairs of semiconductor diodes—three emitting diodes coupled with three recording diodes. Bacterial cells *P. fluorescens* were immobilized on a sensor strip, and the redox electron acceptor 2,6-dichlorophenolindophenol was used as a chromophore. The correlation coefficient with the standard BOD_5_ measurement was high and preserved the value of 0.9 for 6 weeks. Simultaneous detection of 96 channels containing cells can significantly reduce the analysis time and increase the number of simultaneously tested samples [69]. An innovative approach to the detection of environmental toxicity was formulated in [70,71]. The proposed principle of measuring, recording and interpreting data is an absolute game-changer in the study of microbial biological reactions. This approach appears to be even more universal than just a BOD assessment. Its essence is that yeast cells (the yeast *S. cerevisiae* NBRC 0565 is used as an example) are sensitive indicators of biological toxicity assessments (toxicity bioassay). A cell suspension features damped optical oscillations, which can be observed photometrically (by fluorescence intensity or by the content of reduced pyridine nucleotides); the oscillations are stimulated by the addition of glucose. These fluctuations are a function of the state of microorganisms and depend on the concentration of glucose.

The intensity of microbial respiration depends on the content of organic compounds in the analyzed sample, which are oxidized by microorganisms in the presence of oxygen [15,66,72]. The optical-type BOD biosensors are characterized by high sensitivity and stability; besides, they enable the creation of miniaturized sensors [65,72,73]. It is important to note that in the field of developing optical BOD analysis systems, there is also a tendency to use nanomaterials, e.g., nanodiamonds, to improve the characteristics and consumer properties of sensors [72,73,74,75]. Depending on the type of modification and the number of nanodiamonds, the surface of the same polymer may have both bactericidal properties and, conversely, good adhesion to the biomaterial. Precise control of the wettability and biofouling of the surface is achieved by optimizing the conditions of thermal fixation of modified nanodiamonds and allows the creation of sensitive and reliable optical BOD biosensors.

### 3.3. Microbial Fuel Cells as BOD Biosensors

The Japanese researcher I. Karube was also the first to describe a BOD biosensor based on a microbial biofuel cell [76]. Traditionally, the biofuel cell includes an anode anaerobic compartment and a cathode aerobic compartment, which are separated by a proton exchange membrane. In the anode compartment, the biomaterial oxidizes the organic substrate to form carbon dioxide, protons and electrons. Protons pass through the proton-exchange membrane into the cathode compartment, and electrons go to the cathode via an external circuit, generating an electric current. In the cathode compartment, water is formed from protons and oxygen supplied to the system (Figure 13). The electric current generated by the MFC is proportional to the content of organic substrates in the anode compartment, which allows the determination of BOD in the analyzed sample [7].

The works [78,79,80] present MFCs which have a high correlation of BOD measurement results with the standard method (*R*^2^ = 0.95–0.98). A great achievement in the development of BOD sensors was the high long-term stability of the system based on the functioning of a mediator-free fuel cell: the sensor worked for 5 years without any maintenance [81]. At the same time, the disadvantages of this system include a significant response time (about 1 h) and the stationarity of the system, which limits the scope of application of such a BOD sensor. Most of the previously described biofuel cells for BOD analysis had been created on the basis of individual bacterial strains, which somewhat limited the spectrum of substrates decomposed in the MFC [82]. However, recently there has been a tendency to create MFCs based on electroactive biofilms of wastewater microorganisms [83]. Such systems make it possible to achieve oxidation of a wide range of substrates and reach a high correlation with the standard method, but the problem of a long response time does persist [84].

Since BOD electrochemical biosensors based on *Geobacter* (an anaerobic bacterium widely used to develop biosensors and biofuel cells) require anaerobic conditions for anode reactions, they are not used directly in aerobic environments such as aeration tanks. Typically, BOD biosensors are of a closed type, in which the anode is packed inside a closed chamber to avoid exposure to oxygen. In [85], a novel open-type electrochemical biosensor (iBOB) has been developed for on-site monitoring of BOD during intermittent aeration. *Geobacter* spp. exoelectrogenic anaerobes were used to form a bioanode. An open-type anode, without any protection from the effects of oxygen, was directly inserted into a periodic aeration tank filled with sewage from livestock. Figure 14 shows a schematic representation of such a biosensor. It is a membrane-free device which possesses two anodes (consisting of carbon fiber bundles forming a tree-like structure) with a reference electrode and a counter-electrode. In its use, these were placed in an aeration tank. Here, the diffuser for aeration of the medium was placed so that air bubbles from the diffuser prevented the adhesion of suspended solids from wastewater to the reference electrode and ensured their tight adherence to the anode. Due to intermittent aeration, an effective removal of pollutants from the wastewater was achieved. Bacteria present on the outside area of the covered anodes were considered to consume oxygen, by which means anaerobic environments could be produced inside the covered anodes. These results demonstrated that the iBOB biosensor is capable of generating current from the wastewater in the aerating phase.

MFCs require external equipment to evaluate the BOD, which limits their commercialization. In [86], the authors presented a floating biosensor with an autonomous power supply for online monitoring of water quality (Figure 15). To inoculate the anodes, the MFC was placed in a container with pure human urine, equipped with pre-cultured electroactive bacteria adapted for the use of urine as fuel. The biosensor could detect urine in fresh water and turn on visual and audio signals (85 dB). The energy required for the biosensor to operate was produced by the system itself using electroactive microorganisms inside the MFC. When the concentration of urine exceeded the lower threshold corresponding to the concentration of COD 57.7 ± 4.8 mg O_2_ l^–1^, the biosensor triggered an alarm. When switched on, the microbial fuel cell sensor generated a maximum power of 4.3 mW. In the off state, the biosensor produced 25.4 mW. The frequency of the signal was proportional to the concentration of urine. The observed frequencies ranged from 0.01 to 0.59 Hz. This approach made it possible to compare and quantify the presence of water pollution based on the frequency of the signal. The sensor worked autonomously for 5 months. This is the first work about a self-powered device designed for the online monitoring of water quality.

### 3.4. Alternative Approaches to Biosensor Determination of BOD

There are also alternative biosensor approaches to BOD analysis. Thus, a BOD biosensor has been created based on the determination of the concentration of carbon dioxide formed during the biodegradation of organic substances in wastewater [87]. The created analyzer made possible real-time monitoring of measurements of samples taken from treatment facilities. A biosensor based on an array of amperometric sensors and machine learning technologies is described in [88]. This approach allows simultaneous determination of COD, BOD, total organic carbon and other important wastewater indices.

In [89], Fe_3_O_4_ magnetic nanoparticles were used for magnetic labelling of *B. subtilis* cells and their subsequent immobilization on an ultramicroelectrode matrix (UMEA) using an external magnetic field. This modified UMEA was subsequently used as a BOD microsensor. The oxygen consumed is determined amperometrically using UMEA modified by palladium nanoparticles and reduced carboxyl graphene. The design of the developed microsensor is presented in Figure 16. The viability of cells and their fluorescence after the reaction with fluorescein diacetate is preserved. The measurement error is less than 6%. It has been shown that the sensor can be regenerated, and its functioning can be reproduced without any loss of sensitivity; this is a significant difference from biosensors based on the registration of dissolved oxygen and mediator BOD biosensors.

An approach to the creation of a colorimetric (thermistor) BOD biosensor has been described [90], but this trend has not been developed upon.

Apart from the classical BOD assessment procedures, the construction of mathematical models of oxygen consumption by microorganisms during the oxidation of compounds allows building quantitative and qualitative links between occurring processes and phenomena, assessing the contribution of various physical quantities and factors in the formation of the BOD index. An important approach in the BOD assessment methodology is proposed in [91,92]. Two theoretical values are used for the assessment, namely, BOD (biochemical oxygen demand) and *k*_0_ (coefficient of the biochemical oxygen demand rate). They are proposed to be determined using new formulas for two experimental values: BOD_T_ and BOD_2T_ (biochemical oxygen consumption over time periods T and 2T, respectively, expressed in days). The formulation and analytical solution of a direct problem describing the process of biochemical oxidation of organic substances in an aqueous volume in the absence of aeration (e.g., in a reservoir covered with ice or a sealed flask for measuring biochemical oxygen consumption) are given. It is shown how this problem is solved on the basis of a closed system of equations. A closed system, in contrast to the classical Streeter–Phelps system [93], eliminates the possibility of a physically incorrect solution (e.g., a negative concentration of dissolved oxygen). Based on the solution of this direct problem, the inverse problem is formulated, and its solution is given—the formulas for calculating the BOD and *k*_0_. These formulas are used in the compilation of tables for the purpose of illustrating the essence of the proposed method.

## 4. Immobilization Techniques

The method of immobilization of microorganisms is an important determining procedure, because it in fact sets the basic parameters for BOD biosensors. Immobilization determines the lifetime, operational stability, response time, and sensitivity. In this regard, it should be noted that research into the introduction of new immobilization techniques has been constantly under way [36,94]. The most common immobilization methods are adsorption and physical restriction of cells by semipermeable membranes. The best characteristics are shown by biosensors based on microorganisms included in various hydrogels, e.g., an aqueous solution of polyvinyl alcohol [95,96,97], agarose [98], calcium alginate [44], organosilicon sol–gels [95,99] or polycarbomoyl sulfonate [100,101]. Alternatively, it is possible to use disposable sensors based on magnetic nanoparticles, in which the biocomponent can be relatively easily replaced [89]. The formation of biofilms on various surfaces leads to interesting results. For example, in [102] it has been shown that a thinner biofilm of microorganisms is formed on a flame-oxidized stainless-steel anode than on a common carbon cloth anode, which improved the performance of MFCs and leads to faster measurements.

I. Karube, when creating his first microbial BOD biosensor, immobilized microbes on a thin collagen membrane, which was then applied onto a measuring electrode [10]. Electrodeposition of chitosan hydrogel in combination with the joint immobilization of both microorganisms (*S. cerevisiae*) and electron transport mediators is possible [56]. The use of three-dimensional porous graphene materials for the immobilization of microorganisms makes it possible to increase the long-term stability of biosensors [45]. As shown in Figure 17, a significant green fluorescence was observed both on the surface and inside the microbial biofilm of rGO–PPy-B (three-dimensional (3D) porous graphene–polypyrrole (rGO–PPy) composite), which indicates that the immobilized *B. subtilis* cells were viable, and the porous structure was useful for immobilizing more bacteria in a microbial biofilm.

A biosensor based on chitosan cryogel and bovine serum albumin was described in [51]. To obtain a sensor element, a mediator was applied onto the glassy carbon electrode, and after drying it, a layer of cryogel was applied on top. The presence of cryogel in the matrix has a great influence on the structure of the film. Figure 18 shows images of modified electrode surfaces obtained with an electron microscope. Electrodes with a non-cryogel matrix have a flat surface, while those based on a cryogel matrix have shown an interconnected porous structure. This structure increases the surface area, thereby improving the contact between cells and substrates; therefore, the sensitivity of the fabricated biosensor increases.

As polyvinyl alcohol (PVA) has a number of useful properties (non-toxicity, stability, etc.), it is often used to immobilize microorganisms [103]. PVA consists of linear macromolecules, and hydrogels based on it are soluble in water. To create a bioreceptor based on PVA, the hydrogel must have a three-dimensional network structure, which can be provided by crosslinking agents [104]. For example, in [96], *Paracoccus yeei* bacteria were immobilized in a matrix of PVA modified with N-vinylpyrrolidone. The developed hydrogel formed a network structure (Figure 19) which made it possible to capture microbial cells while preserving their viability and biocatalytic properties, and to provide the technological possibility of replicating disposable receptor elements of the biosensor.

PVA or other polyalcohols can be used not only as a chemically modified polymer, but also as a structure-controlling agent in a silicon sol–gel matrix [15]. This approach makes it possible to create a composite material characterized by insolubility in water, while maintaining non-toxicity and biocompatibility. It is known that silicas based on alkoxysilanes are inert, non-toxic, biocompatible and do not dissolve in water, and the synthesis of materials based on them can be carried out under mild conditions using sol–gel technologies [105]. When using a sol–gel technology, easier and simpler formation of electrochemical and optical BOD biosensors is possible [15,99]. The immobilization of microorganisms into organosilicate matrices allows each cell to be encapsulated individually, preventing its further division [25]. By measuring the oxidative activity of a BOD biosensor based on *D. hansenii* yeast cells, it has been shown in [25] that the shells around microbial cells do not interfere with the diffusion of the substrate and have excellent properties, such as protection from extreme environmental factors (heavy metal ions, high pH, UV radiation).

## 5. Parameters of BOD Biosensors

The efficiency of BOD measurement using a biosensor analyzer is influenced by its analytical and metrological characteristics, which include sensitivity, selectivity, repeatability, reproducibility, analysis time and stability of the receptor element. Optimization of the work of a biosensor system is a complex task, because actions to improve one characteristic can negatively affect another.

Two main approaches are used to determine the analytical signal of a biosensor analyzer. The first is based on determining the difference between the signals of the biosensor transducer before and after the introduction of the sample in a state of equilibrium. If this approach is used, the analysis time can range from several minutes to several hours. Another approach is based on the analysis of the kinetic dependence of the sensor response value on time. The kinetic dependence is characterized by a signal that is measured after the sample is added and is associated with the stimulation of microbial metabolism intensity. In this case, the analysis time can range from a few seconds to several minutes. Besides, this approach makes it possible to achieve a wider range of detectable BOD values. However, it may lead to a slight loss of reproducibility of the analysis [106]. The kinetic approach is preferable to the equilibrium approach in cases where it is important to analyze a large number of samples.

To conduct a quantitative analysis, it is necessary to know the dependence of the biosensor signal on the level of biochemical consumption of the sample. The linear section of this dependence determines the range of analyzed BOD values in the sample. A wide range of the linear dependence simplifies the analysis and reduces the error due to the absence of sample dilution. Besides the range of detectable BOD values, the sensitivity characteristics such as the sensitivity coefficient and detection limit are used to compare the effectiveness of biosensors. The sensitivity coefficient is the dependence of the value of the derivative of biosensor’s analytical signal on the BOD value. The detection limit is the smallest value of the BOD value that can be detected in a sample. As a rule, it is possible to increase the sensitivity of the biosensor by increasing the number of microorganisms in the receptor. Besides, the described characteristics are influenced by the type of transducer and the method of immobilization of the biomaterial on the transducer. BOD biosensors with a high content of microorganisms in the receptor element are usually more sensitive but are characterized by a narrow linear range. These characteristics are also affected by the sensitivity of the biosensor to certain organic substances. The BOD biosensor may have different linear characteristics when using different calibration standards and samples with the different compositions of organic substrates. As biosensors BOD analyzers have been created for the development of a rapid analytical method, and the measurement with their help should be no less accurate than the classical BOD_5_ method. The repeatability of BOD biosensor analyzers is most often within the range of 10–11% for biosensors based on a single microbial strain and can increase up to 15% when using associations of microorganisms [7].

The presence of surfactants in wastewater can lead to the damage of the biosensor receptor, therefore, when developing a biosensor, the substrate specificity of the biosensor is verified, inhibitors and activators are identified, the operational and long-term stability of the biosensor is evaluated, and an assessment is made of how individual reagents from wastewater affect these characteristics.

Recently, new models of BOD biosensors have been created based on the transducers described above [35,36]. In these instances, researchers pay special attention to improving the characteristics of the analyzers—increasing the stability and sensitivity of the analysis, increasing the correlation of the results obtained by the standard method and using biosensors. First of all, the aim is to search for new microorganisms, to use modern materials and new methods of immobilization of microorganisms. The main parameters of various types of BOD biosensors presented in scientific publications are given in more detail in Table 1.

Thus, to date, models of BOD biosensors based on a wide range of converter types are known. It can be noted that, in recent years, the number of works using methods of direct oxygen detection (Clark electrode or optical oxygen sensor) has significantly decreased. At the same time, a significant increase of works on the use of MFC for BOD analysis is observed, since such systems are characterized by a high correlation of the analysis results with the standard method, and a very high stability. A common problem of many of the presented biosensor systems is a sufficiently high lower limit of the determined BOD_5_ values, which does not allow such systems to be used in the analysis of surface water samples in which BOD_5_ is usually within the range of 1 to 4 mg/dm^3^.

## 6. Problems and Achievements

### 6.1. Standard Solution for Calibration of Biological Devices

An important step in assessing the correct functioning of the BOD biosensor receptor system is its calibration. The use of a suitable standard solution is one of the factors providing a high correlation between the BOD values determined using the biosensor technique and the standard method. Most often, a glucose–glutamate mixture is used as a calibration solution (GGA, a mixture of glucose and glutamic acid with concentrations of 150 mg/dm^3^, corresponding to 205 mg O_2_/dm^3^ BOD_5_). Despite the widespread use of GGA in the standard method of BOD analysis [4,133,134], it has a number of disadvantages in the calibration of BOD biosensors. Thus, GGA stores poorly and is easily exposed to microbial infection. Besides, glutamic acid, which is part of it, is more slowly oxidized by microbes in the presence of glucose. This has almost no effect on the results of the standard method due to its duration but may affect the results of the BOD biosensor measurement. Another disadvantage of GGA is that it consists of two easily oxidizable substances, whereas substances with low oxidation rates are mainly contained in wastewater and surface waters [7]. For these reasons, one of the directions of the development of BOD biosensors is the creation of calibration mixtures similar in composition to real water samples [20,131,135,136,137,138,139]. One example of such a mixture are synthetic wastewaters developed by the Organization for Economic Cooperation and Development (OECD). The components of this standard are meat extract, peptone, urea and inorganic salts [140]. The use of this calibration standard can increase the correlation between the results of the standard method and the biosensor data [33].

In MFC-based biosensors, the BOD level is estimated based on the mono-type dependence between the generated current and the actual content of biodegradable organic substances in the anolyte [141]. However, the result of this measurement method applies only to soluble easily biodegradable contents, because the slowly biodegradable suspended organic fraction of real wastewaters takes a considerable time (from several hours to several days) to be hydrolyzed and to biodegrade [142]. To solve this problem related to the calibration of a device under development, additional research is required.

### 6.2. Correlation of Results of Biological Devices and Standard Methods

It should be noted that the value of the BOD index, determined using biosensors, does not always correspond to the results of the standard 5-day method. This is due to the fact that, over a short time, the biomaterial in the biosensor’s receptor element is not capable of oxidizing the same wide range of substances as activated sludge in 5 days. This problem is especially aggravated when there is a large amount of slow-oxidizing organic substances in the sample [7]. Therefore, research of the recent years has been aimed at increasing the correlation of the results of the rapid BOD analysis with the standard method. As a rule, the correlation coefficient of these results is within the range of 0.9–0.99 [7,36]. The main ways to increase the correlation are the use of microorganisms with a wide range of oxidizable substances [98,118] and their associations [13,16]; adaptation and increased stability in biosensor systems of activated sludge and other natural communities of microorganisms [112,119]; the use of specialized standards for biosensor calibration, as mentioned above [23].

Another interesting approach is presented in [143]—the use of a set of three MFCs connected hydraulically in series (a schematic diagram of the device is shown in Figure 20). The sum of the current density generated by each MFC in the array was calibrated according to BOD_5_, and a linear characteristic up to 720 mg l^−1^ BOD_5_ (1175 mg l^−1^ COD) was obtained. The dynamic range was twice as large as that of the first MFC in the array, operating on a separate basis. The average response time to achieve a stable current for reliable quantification was 2.3 h. The anode biofilm of the sensor was dominated by Geobacter spp. and representatives of Porphyromonadaceae. The dynamic range of the multi-stage MFC sensor could be increased by increasing the hydraulic holding time. The modular mode of operation demonstrated here makes it possible to expand the dynamic range of the sensor, allowing the analysis of BOD concentrations within the range from typical municipal wastewaters to those found in some industrial wastewaters. It is important to note that with this practical and inexpensive MFC configuration, it is possible to provide an accurate measurement of both low and high levels of organic carbon without the need for corrections or dilution online and in real time.

### 6.3. Commercialization of BOD Biosensors and MFCs

Many studies are devoted to the creation of biosensors for the analysis of BOD in strictly defined types of real samples rather than universal analyzers [44,127]. The demand for such studies has led to the commercialization of the most successful models of BOD biosensors [7,35,36].

It should be noted that most BOD biosensor analyzers described in scientific publications are still at the stage of laboratory models. However, there are currently several commercial rapid BOD analyzers on the market. The first such analyzer was released by Nisshin Electric Co. Ltd. (Saitama, Japan) in 1983. This method of analyzing the biochemical oxygen demand using a biosensor analyzer was included in the Japanese Industrial Standard (JIS) only in 1990 (JIS K3602). After that, commercial BOD biosensors have been released to the market by other European (Dr Lange GmbH (Hamburg, Germany), Aucoteam GmbH (Berlin, Germany), Prufgeratewerk Medingen GmbH (Ottendorf-Okrilla, Germany) and Japanese manufacturers (Central Kagaku Corp., Tokyo Japan). The first commercialized BOD analyzers were biofilm-type biosensors based on the oxygen electrode. As a biomaterial in the first models, as a rule, activated sludge was used [7]. Currently produced commercial BOD analyzers are most often based on the principle of a bioreactor (Table 2).

Commercially available BOD analyzers differ by the method of introducing biomaterial into the measuring cell (reactor). Thus, active sludge microorganisms are constantly introduced into the Ra-BOD analyzer. In the Biox-1010 analyzer, the biomaterial used is seeded into small plastic cells located inside the biological reactor. This is done to increase the contact area of the biomaterial and protect it from the mechanical impact during the mixing. In the MB-DBO analyzer, the biomaterial is grown in a separate, independent compartment of the bioreactor. This makes it possible to create a stable population of microorganisms and to use the main measuring compartment of the bioreactor continuously. The QuickBOD α1000 analyzer is one of the few that are based on the use of a pure culture of the yeast microorganism *Trichosporon cutaneum*. The BioMonitor analyzer features an unusual design—it consists of four bioreactors connected in series. Herewith, the entire system works on the principle of an aerotank (Figure 21). Oxygen demand in biochemical reactions is measured by an oxygen sensor mounted in the last bioreactor. The BOD calculation is based on the difference between the measured oxygen consumption in the analyzed sample and the control sample with activated sludge, but without a sample of the analyzed water. Despite the complexity of the design, this approach makes it possible to increase the degree of utilization of organic substances in the sample and reduce the analysis time to 3–4 min.

An important advantage of commercial BOD biosensors is the speed of analysis. Unfortunately, most of the commercial BOD biosensor analyzers are designed according to the principle of a bioreactor, so they are stationary and have a large mass and dimensions (weight up to 200 kg) (Figure 22 shows photos of some commercially produced BOD biosensor analyzers). Besides that, BOD biosensors provide reliable and reproducible results under strictly defined conditions. For example, the physico-chemical characteristics of the studied water samples should be approximately on the same level. Besides, an important factor in obtaining the correct results is the selection of calibration mixtures that have similarities in composition with the test samples. The biomaterial used in the biosensor must be adapted to the test samples and calibration solutions. Thus, BOD biosensor analyzers are most effective for continuous monitoring of the same samples with a similar composition. When determining the BOD of samples with different compositions and from different sources, biosensor analyzers can make measurements with a large error [7,35,36].

An important issue of the active commercialization of BOD analyzers is the absence of or gaps in regulatory documentation in a number of countries. Certification of a new technique or a new device in different countries with different legislation can be a long and time-consuming process. Besides the delivery of the analyzers themselves, the manufacturer must establish a regular supply of biological material, which can also cause complications when crossing borders.

## 7. Conclusions

Based on the presented data, the analysis of the biochemical oxygen demand using biosensors is a promising area of analytical biotechnology. BOD biosensors serve as reliable analytical tools that can be used for continuous monitoring of aquatic ecosystems and wastewaters. The development of BOD sensors has come a long way from simple systems based on the Clark oxygen electrode to systems based on direct electron transfer using microbial fuel cells. Modern trends in the development of biosensors, such as the active use of nanomaterials, electroactive biofilms and machine learning technologies for signal processing, have made it possible to increase the sensitivity of BOD analysis. The use of mediator screen-printed electrodes or MFCs has led to a reduction in the size of the sensors themselves and their cost. MFCs for BOD analysis have come from mediator to mediator-less devices [18], with the design changing from two-chamber to single-chamber [142,145]. These advances contributed to greater accuracy and reproducibility of BOD measurements to enable using these developments in the commercialization of laboratory models. Detailed studies have been carried out to improve the accuracy and reproducibility of sensors for commercial use. Nevertheless, the search continues for suitable microorganisms, approaches to their immobilization and ways to obtain an analytical signal. Works on autonomously operating biosensors and MFCs for the determination of BOD, which is perhaps the most promising direction of development today, are being published. Modern approaches to the development of biosensors and biofuel cells are associated with the active use of artificial intelligence (AI) [146,147]. These devices combine both wireless technologies and advanced machine learning algorithms with expanded diagnostic and data processing functions, ultimately leading to the decision-making level upon the analysis of the data obtained. In the future, such sensors can be used in remote and hard-to-reach areas to monitor water purity. Herewith, such devices will be able not only to signal about water quality, but also to start and stop additional wastewater treatment processes as well as to make decisions about the possibility of these waters getting into the world ocean. In the long run, such systems based on MFC and AI can be used as alternative energy sources of autonomous action.

## Figures and Tables

**Figure 1 biosensors-12-00842-f001:**
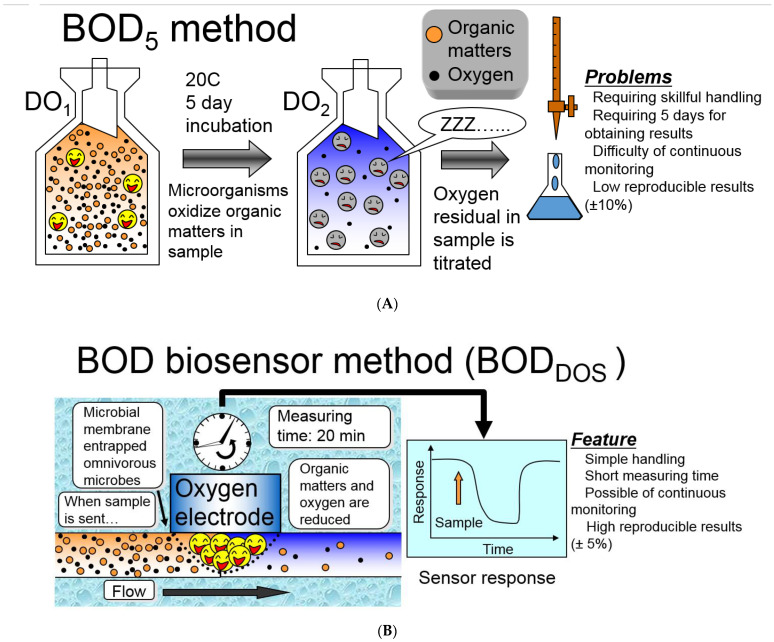
(**A**) Schematic diagram of the BOD classical analysis [6]. Reprinted with permission from ref. [6]. Copyright © 2022 American Scientific Publishers; (**B**) schematic diagram of an amperometric biosensor.

**Figure 2 biosensors-12-00842-f002:**
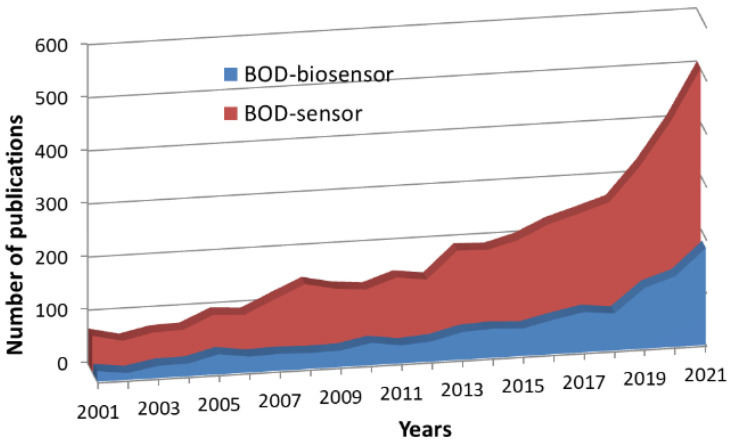
Number of publications in the ScienceDirect database per year by the BOD biosensor and BOD sensor keywords.

**Figure 3 biosensors-12-00842-f003:**
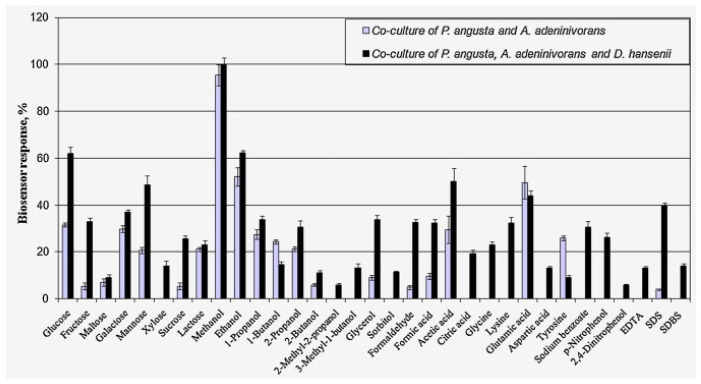
Substrate specificities of receptor elements based on the co-cultures used. The change in biosensor’s receptor element substrate specificity upon addition of a new microorganism into the receptor is shown [13]. Reprinted with permission from ref. [13]. Copyright © 2022 Elsevier.

**Figure 4 biosensors-12-00842-f004:**
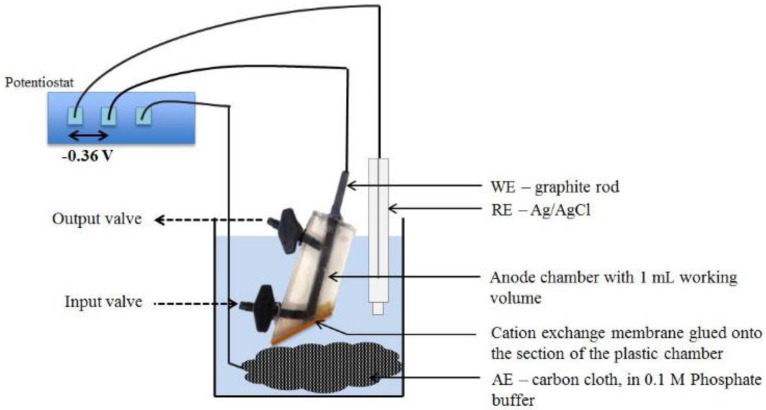
Three-electrode configuration of an electroactive film-based MFC reactor: WE, working electrode; RE, reference electrode; AE, auxiliary electrode [32]. Reprinted with permission from ref. [32]. Copyright© 2022 Elsevier.

**Figure 5 biosensors-12-00842-f005:**
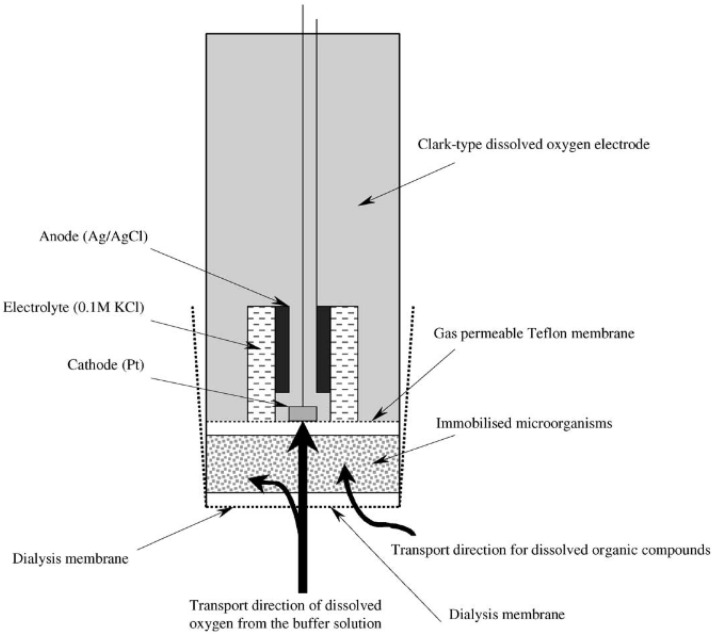
Schematic representation of a BOD biosensor. An immobilized microbial population in combination with a Clark-type oxygen electrode [33]. Reprinted with permission from ref. [33]. Copyright© 2022 Elsevier.

**Figure 6 biosensors-12-00842-f006:**
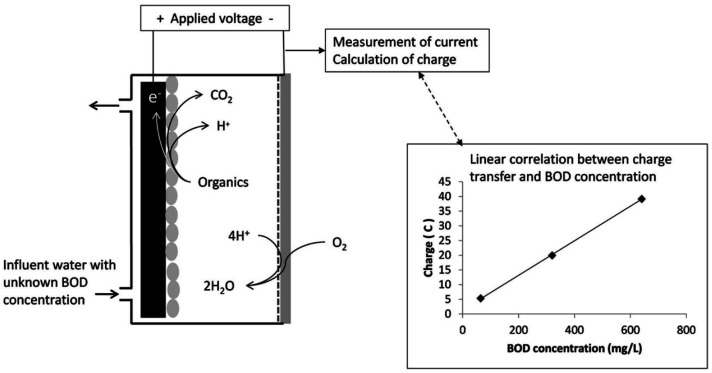
Schematic of a bioelectrochemical reactor used as a BOD sensor [34]. Reprinted with permission from ref. [34]. Copyright© 2022 Elsevier.

**Figure 7 biosensors-12-00842-f007:**
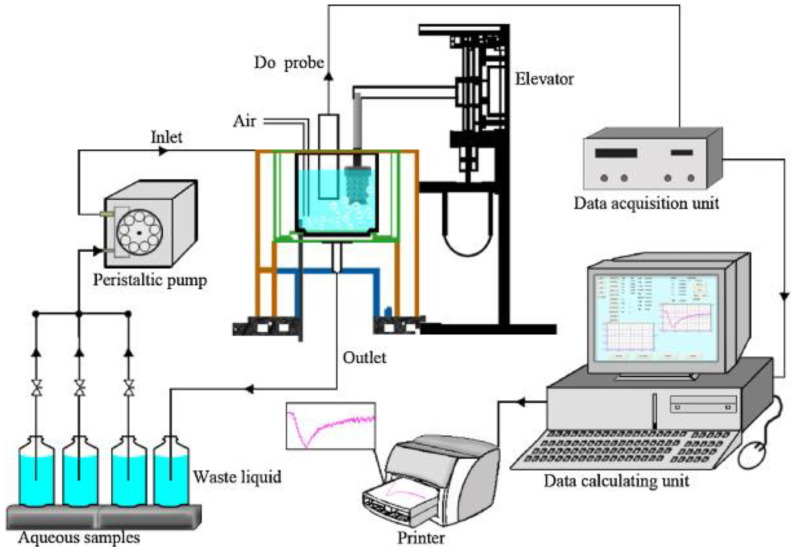
Schematic process of a reactor-type BOD biosensor system [37]. Reprinted with permission from ref. [37]. Copyright© 2022 Elsevier.

**Figure 8 biosensors-12-00842-f008:**
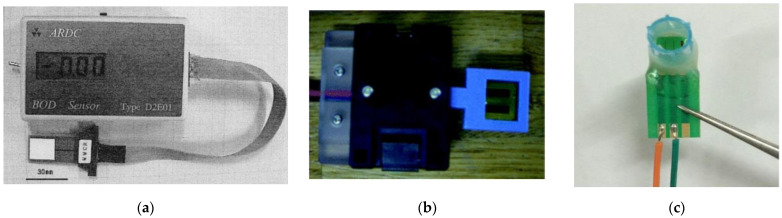
Portable biosensor BOD analyzers. (**a**) Biosensor with a disposable receptor element based on *P. fluorescens* [40]. The size of the device, which has a built-in microcomputer, is 170 × 80 × 30 mm^3^. Reprinted with permission from ref. [40]. Copyright© 2022 Elsevier. (**b**) Biosensor based on a disposable microbial (*S. cerevisiae*) electrode chip (inner volume, 563 mL); the design included a micro-stirrer system [41]. Reprinted with permission from ref. [41]. Copyright© 2022 Royal Society of Chemistry. (**c**) Microsensor based on poly(neutral red) and *P. aeruginosa* bacteria modified interdigitated ultramicroelectrode array (IUDA) [42]. The volume of the measuring chamber was 1 mL; sensing area, 5 mm^2^. Reprinted from ref. [42]. Copyright© 2022 Creative Commons Attribution 4.0 International License.

**Figure 9 biosensors-12-00842-f009:**
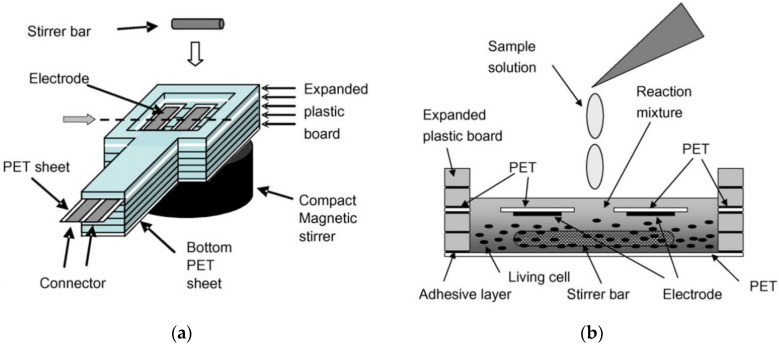
Schematic diagram of a sensor device. (**a**) Squint view of the sensor device. (**b**) Cross-sectional view from a gray arrow in the diagram of the squint view [21]. PET, polyethylene terephthalate. Reprinted with permission from ref. [21]. Copyright© 2022 Elsevier.

**Figure 10 biosensors-12-00842-f010:**
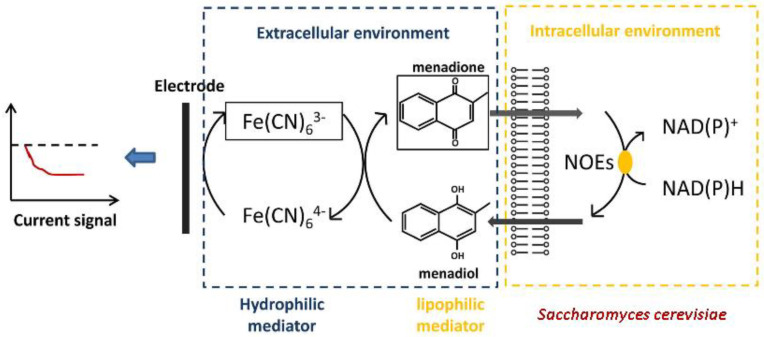
Mechanism of operation of a two-mediator system of potassium hexacyanoferrate (III)–menadione with *S. cerevisiae* cells [56]. Reprinted with permission from ref. [56]. Copyright© 2022 Elsevier.

**Figure 11 biosensors-12-00842-f011:**
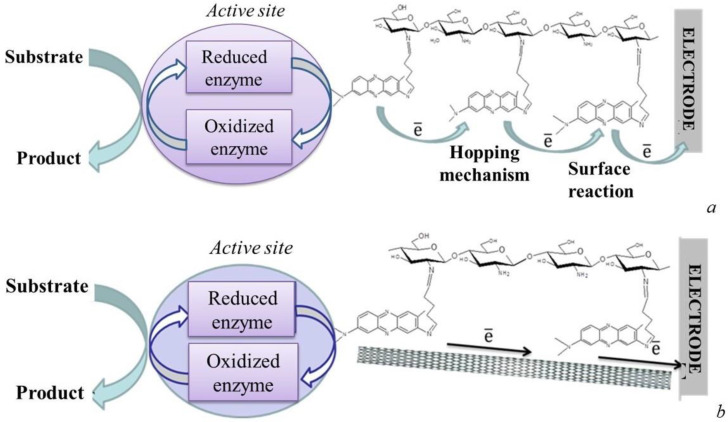
Transfer of electrons to the electrode in a chitosan–neutral red redox polymer: (**a**) in the absence of carbon nanotubes; (**b**) in the presence of carbon nanotubes [57]. Reprinted with permission from ref. [57]. Copyright© 2022 Elsevier.

**Figure 12 biosensors-12-00842-f012:**
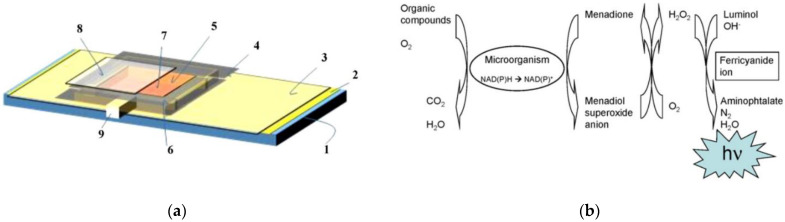
(**a**) Schematic design of a BOD biosensor chip; the components are (1) SO_3_ glass slide, (2) Ru-complex dye–oxygen sensing film, (3) polyethylene–polypropylene film, (4) 0.5 mm thick silicone rubber sheet, (5) biofilm, (6) 1 mm thick silicone rubber sheet, (7) sample cavity or sample IN, (8) cover glass, (9) sample OUT [66]. Reprinted with permission from ref. [66]. Copyright© 2022 Elsevier. (**b**) Principle of the BOD_chemiluminesc_ method for BOD measurements. Not in stoichiometry [67]. Reprinted with permission from ref. [67]. Copyright© 2022 Elsevier.

**Figure 13 biosensors-12-00842-f013:**
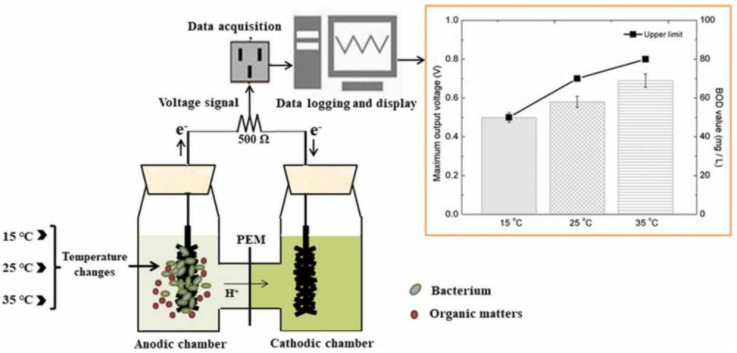
Schematic of an MFC-based BOD biosensor [77]. Reprinted with permission from ref. [77]. Copyright© 2022 Elsevier.

**Figure 14 biosensors-12-00842-f014:**
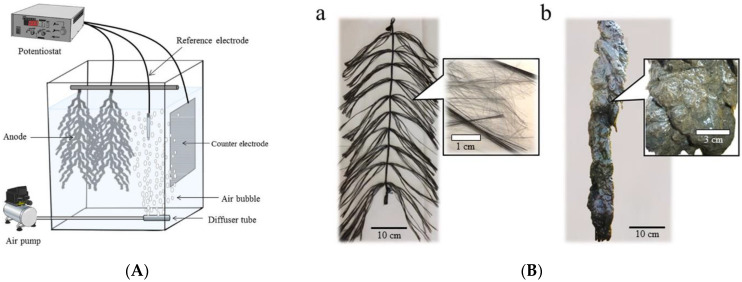
(**A**) Schematic representation of the iBOB biosensor inserted into an intermittently aerating tank. Reprinted from ref. [85]. Copyright© 2022 Creative Commons Attribution 4.0 International license. (**B**) Anodes of the iBOB biosensor before (**a**) and after (**b**) use. The anode completely covered with suspended solids, including hay feed, is shown [85].

**Figure 15 biosensors-12-00842-f015:**
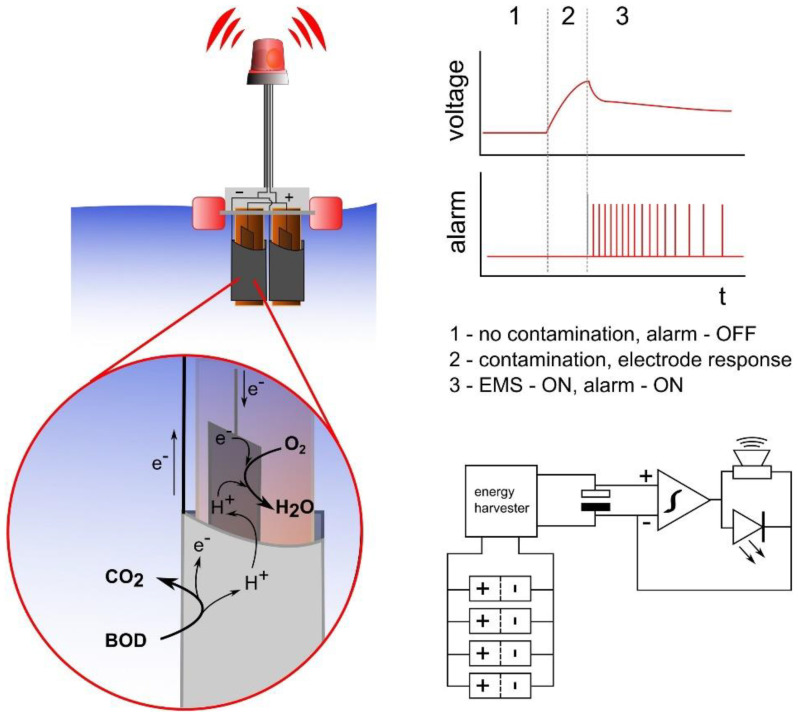
Schematic representation of the biosensor and the principle of operation. 1, biosensor operates in uncontaminated freshwater under open-circuit conditions; 2, in the presence of urine, the sensor open-circuit voltage increases; 3, the energy management system (EMS) switches ON, resulting in the charging of the capacitor up to a threshold; the audio and visual alarm is activated by the capacitor when full, causing the latter to discharge. The system is able to repeatedly charge/discharge the capacitor [86]. Reprinted with permission from ref. [86]. Copyright© 2022 Elsevier.

**Figure 16 biosensors-12-00842-f016:**
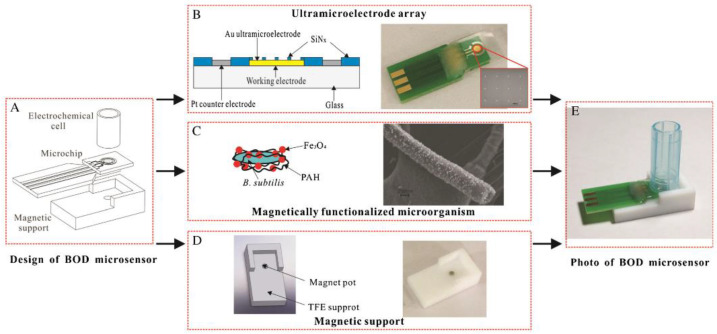
Design of a BOD microsensor (**A**); the structure sketch and photo of UMEA (**B**); structure sketch and SEM images of Fe_3_O_4_-functionalized *B. subtilis* (**C**); design and photo of tetrafluoroethylene support (**D**); photo of the sensor setup (**E**) [89]. Reprinted from ref. [89]. Copyright© 2022 Creative Commons Attribution 4.0 International license.

**Figure 17 biosensors-12-00842-f017:**
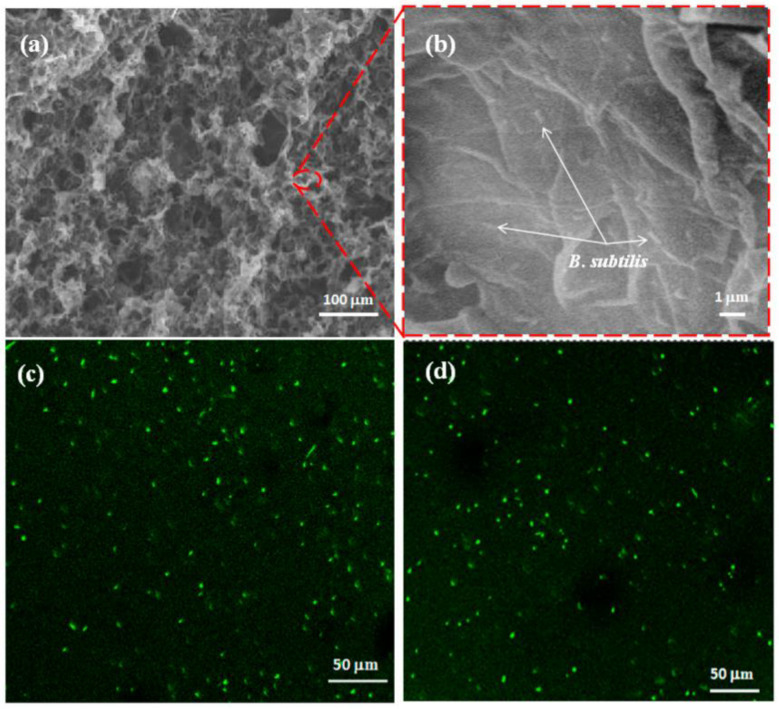
SEM and confocal laser scanning microscopy (CLSM) images of rGO–PPy-B microbial biofilm. (**a**) Typical SEM image of the interior microstructure of rGO–PPy-B microbial biofilm; (**b**) the magnified SEM image of (**a**); CLSM images of rGO–PPy-B microbial biofilm collected at the surface (**c**) and 6 mm depth (**d**) [45]. Reprinted from ref. [45]. Copyright© 2022 Creative Commons Attribution 4.0 License.

**Figure 18 biosensors-12-00842-f018:**
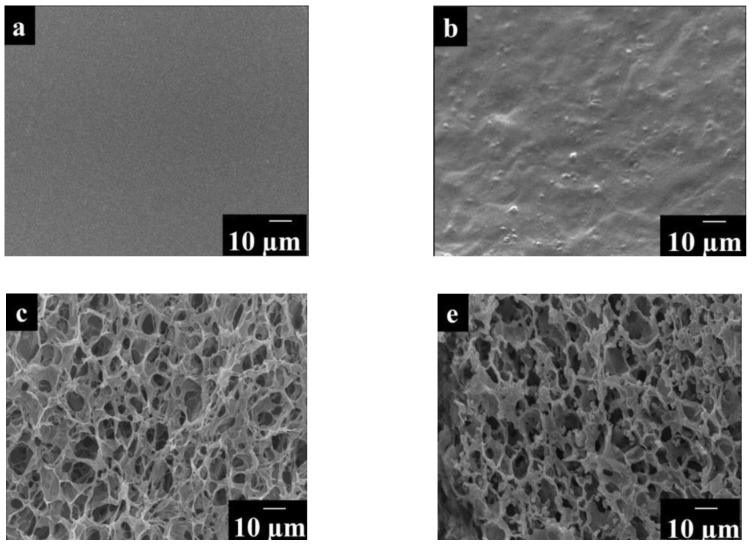
Electron micrographs of (**a**) chitosan–ex-cryogel matrix BSA; (**b**) chitosan–BSA–ex-cryogel matrix-activated sludge; (**c**) chitosan–cryogel matrix BSA; (**e**) chitosan–BSA–cryogel matrix activated sludge [51]. Reprinted with permission from ref. [51]. Copyright© 2022 Elsevier.

**Figure 19 biosensors-12-00842-f019:**
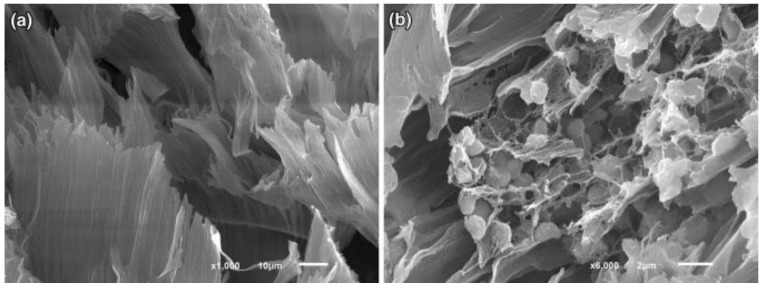
Scanning electron micrographs of N-vinylpyrrolidone-modified PVA. (**a**) A modified PVA matrix after swelling in a buffer solution; (**b**) a modified PVA matrix with immobilized *P. yeei* bacteria after swelling in a buffer solution [96]. Reprinted from ref. [96]. Copyright© 2022 Creative Commons Attribution 4.0 International license.

**Figure 20 biosensors-12-00842-f020:**
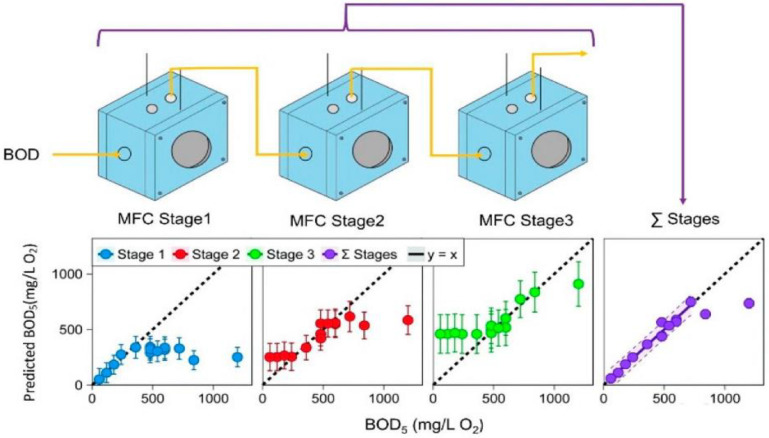
Schematic diagram of the three-stage MFCs as BOD sensor and compliance of predicted BOD_5_ values with five-day BOD test (BOD_5_). *y* = *x* is shown as the “ideal” prediction [143]. Reprinted with permission from ref. [143]. Copyright© 2022 Royal Society of Chemistry.

**Figure 21 biosensors-12-00842-f021:**
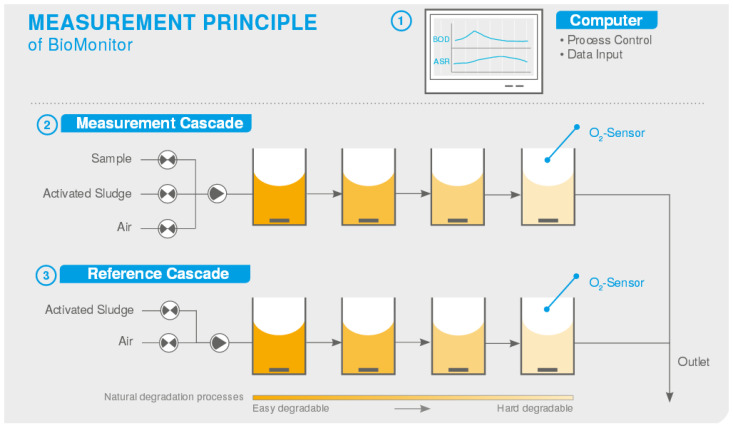
Schematic diagram of a BioMonitor BOD analyzer (LAR, USA) [144]. (1) Process control, measurement results display, interface diagram of periphery analyzers; (2) the aerated mixture of activated sludge and sample is pumped through the four reactors. The O_2_ sensor determines the oxygen consumed during decomposition; (3) the activated sludge is aerated and its respiration is determined (ASR).

**Figure 22 biosensors-12-00842-f022:**
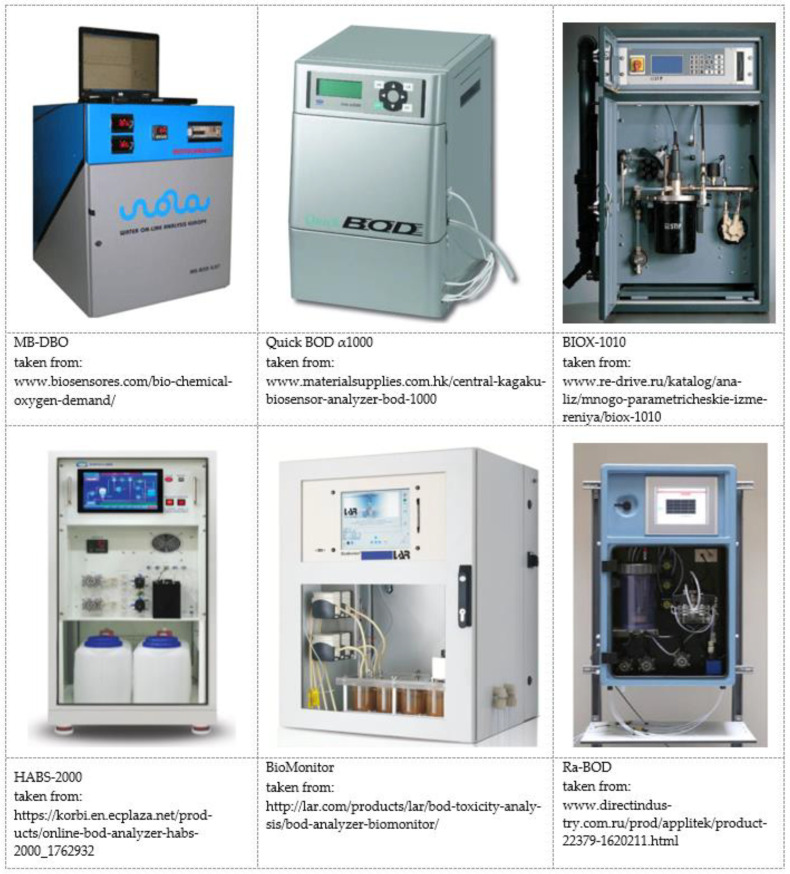
The main models of commercially produced BOD biosensor analyzers (photos are presented on the manufacturers’ web pages).

**Table 1 biosensors-12-00842-t001:** Characteristics and parameters of BOD sensors presented in scientific publications.

Microorganisms, Immobilization	Measured Samples	Major Characteristics	References, Year
**Oxygen Electrode-Based Amperometric BOD Biosensors**			
A preparation of activated sludge (BOD_seed_), the cells in which are killed by heating at 300 °C for 1.75 min. Immobilization by adsorption onto a polycarbonate membrane	Glucose glutamic acid (GGA) mix/wastewaters	Range of measured BOD_5_ values, up to 40 mg O_2_/dm^3^; measurement time, 40–45 min; sensitivity, 0.412 μA/(mg BOD /L)	[107], 2005
Association of microorganisms immobilized by adsorption on a nylon membrane	GGA/wastewaters	Response time, 90 min; long-term stability, 60 days; lower detection limit, 1 mg O_2_ mg O_2_/dm^3^; range of measured BOD_5_ values, up to 75 mg O_2_/L; correlation between BOD_biosens_ and BOD_5_ (deviation, 10%), convergence, 3.49–4.45%; sensitivity, 2.8 nA/(mg BOD/L)	[9], 2008
*P. putida* bacteria immobilized in agarose gel	GGA/synthetic wastewaters (OECD). Effluents with increased content of phenol	Range of measured BOD_5_ values, 0–50 mg O_2_/dm^3^; measurement time, 15–45 min; sensitivity, 0.0136 n.r./(mg BOD/L)	[108], 2008
*S. cerevisiae* yeast immobilized in calcium alginate	GGA/wastewaters	Response time, 90 min; R = 0.95; long-term stability, 30 days; range of measured BOD_5_ values, 2–30 mg O_2_/dm^3^; sensitivity, 0.06 ppm of dissolved oxygen/(mg BOD/L)	[109], 2009
Commercial preparation of microorganisms of activated sludge (BOD_seed_). Immobilization into an organosilicon sol–gel based on silicic acid and grafted copolymer of polyvinyl alcohol and 4-vinylpyridine	GGA/ synthetic wastewaters (OECD)	Lower boundary of detected BOD_5_ values, 0.5 mg O_2_/dm^3^; measurement time, 10 min; long-term stability, 50 days; *R* = 0.9851; range of measured BOD_5_ values, up to 50 mg O_2_/dm^3^; sensitivity, 19.7 mV/(mg BOD_5_/L)	[95], 2011
Microorganisms of *D. hansenii* VKM Y-2482. Immobilization using a dialysis membrane	GGA/wastewaters of treatment plants and glucose treacle plant	Range of measured BOD_5_ values, 2.2–177 mg O_2_/dm^3^; measurement time, 10–17 min; reproducibility, 4%; long-term stability, >30 days; *R* = 0.98; sensitivity, 0.21 nA dm^3^/min mg	[17], 2012
Biofilm of a natural community of microorganisms from local wastewaters grown on a glass substrate	GGA, synthetic wastewaters (OECD). Wastewaters of treatment plants, food industry and surface waters	Range of measured BOD_5_ values, 0.5–30 mg O_2_/dm^3^; measurement time, 6–8 min; reproducibility, 3.9%; long-term stability, 17 months; *R* = 0.983; sensitivity, 14.7 nA/(mg GGA/L)	[110], 2012
*Ps. fluorescens* P75 bacteria immobilized in agarose gel	Synthetic wastewaters (OECD), wastewaters of meat processing plants	Linear range of measured BOD_7_ values, 5–40 mg O_2_/dm^3^; long-term stability, 80 days; measurement time, up to 20 min	[98], 2012
*Aeromonas hydrophila* P69.1 bacteria immobilized in agarose gel	Synthetic wastewaters (OECD), wastewaters of meat processing plants	Linear range of measured BOD_7_ values, 5–45 mg/dm^3^; long-term stability, 90 days; measurement time, up to 20 min; sensitivity, 0.019 n.r./(mg BOD_7_/L)	[98], 2012
Yeast *D. hansenii* immobilized in poly(vinyl alcohol) modified by N-vinylpyrrolidone	GGA, wastewaters of treatment plants and food productions	Range of measured BOD_5_ values, 0.7–206.7 mg O_2_/dm^3^; measurement time, 5–7 min; reproducibility, 4.2%; long-term stability, 30 days; *R*^2^ = 0.9911; sensitivity 0.0045 min^−1^	[22], 2013
Individual biosensors based on *P. fluorescens*, *A. hydrophila*, *P. putida*, *E. coli*, *B. subtilis*, *Paenibacillus* sp., *Microbacterium phyllosphaerae* bacteria, immobilized in agarose gel. Mathematical processing of the data of 7 sensors for accurate determination of BOD	Synthetic wastewaters (OECD) and waters modelling the composition of effluents of meat, milk, oil shale and paper industries	Standard deviation of the results, less than 3.4%	[23], 2013
*P. putida* bacteria immobilized on a nitrocellulose membrane	GGA/synthetic wastewaters. River water samples	Range of measured BOD_5_ values, 0.5–10 mg O_2_/dm^3^; measurement time, 15 min; long-term stability, 10 days; sensitivity, 0.175 μA/(mg O_2_/L)	[111], 2013
Biofilm of microorganisms from wastewater treatment plants on a hydrogel of reduced graphene oxide	GGA/synthetic wastewaters (OECD)	Range of measured BOD_5_ values, 2–64 mg O_2_/dm^3^; BOD_5_ detection limit, 0.4 mg O_2_/dm^3^; measurement time, 10 min; long-term stability, 2 months; *R* = 0.981; sensitivity, 4.8 nA/(mg O_2_/L)	[112], 2015
Co-cultures of the yeasts *Pichia angusta*, *A. adeninivorans* and *D. hansenii* immobilized in N-vinylpyrrolidone-modified poly(vinyl alcohol)	GGA/river and swamp waters, sewage from urban sewage facilities, meltwater, fermentation products of starchy raw materials (distillery stillage)	Range of measured BOD_5_ values, 2.4–80 mg O_2_/dm^3^; measurement time, 3 min; reproducibility, 8.9%; long-term stability, 17 days; *R*^2^ = 0.9988; sensitivity coefficient, 10 s^−1^ × 10^−5^	[13], 2015
Biofilm of wastewater microorganisms. Biosensor of bioreactor type	GGA/samples of surface waters	Long-term stability, over 30 days	[38], 2016
*E. coli* bacteria immobilized in collagen fiber with Zr(IV)	GGA/synthetic wastewaters (OECD). Samples of river water	Range of measured BOD_5_ values, up to 225 mg O_2_/dm^3^; long-term stability, 42 days; sensitivity 0.0085 (mg/L of dissolved oxygen)/(mg BOD/L)	[26], 2017
Yeast *D. hansenii* in an organosilicate matrix based on polyethylene glycol, tetraethoxysilane and methyltriethoxysilane	GGA/ samples of river and well water, as well as sewage and meltwater	Range of measured BOD_5_ values, 0.5–5 mg O_2_/dm^3^; reproducibility, 6.7%; long-term stability, 40 days; *R*^2^ = 0.9988; sensitivity coefficient, 0.40 nA L/mg O_2_	[113], 2018
*P. yeei* VKM B-3302 bacteria isolated from activated sludge and immobilized in an N-vinylpyrrolidone-modified poly(vinyl alcohol) matrix	GGA/wastewaters of municipal treatment facilities, effluents of food industry plants, natural waters, including those of ponds polluted with fuels and lubricants	Range of measured BOD_5_ values, 0.05–5.0 mg O_2_/dm^3^; reproducibility, 7%; measurement time, 4–6 min; long-term stability, 45 days; *R*^2^ = 0.9990; sensitivity coefficient, 110 10^−5^ × s^−1^	[96], 2020
Activated sludge bacteria *P. yeei*, *P. veronii* and *B. proteolyticus* immobilized layer-by-layer in a hydrogel of polyvinyl alcohol modified with N-vinylpyrrolidone	GGA/ wastewaters of municipal treatment facilities, effluents of food industry plant, waters from ponds and rivers within the municipal area	Range of measured BOD_5_ values, 0.51–3.80 mg O_2_/dm^3^; measurement time, 5–12 min; reproducibility, 9.5%; long-term stability, 52 days; *R* = 0.9956; sensitivity coefficient, 120 10^−5^ × s^−1^	[97], 2021
Activated sludge bacteria *P. yeei* in an organosilica sol–gel matrix consisting of tetraethoxysilane, methyltriethoxysilane and polyvinyl alcohol as a structure-modifying agent	GGA/ samples of natural surface waters and industrial wastewaters	Range of measured BOD_5_ values, 0.01–20 mg O_2_/dm^3^; measurement time, 4–6 min; reproducibility, 3%; long-term stability, 31 days; *R* = 0.9983; sensitivity coefficient, 30 10^−3^ × min^−1^	[27], 2021
Mixture of the bacteria *P. yeei* VKM B-3302 and the yeast *D. hansenii* VKM Y-2482 in a tetraethoxysilane / methyltriethoxysilane sol–gel matrix	GGA/surface waters of rivers and ponds	Range of measured BOD_5_ values, 0.5–2.9 mg O_2_/dm^3^; reproducibility, 3%; long-term stability, 42 days; *R*^2^ = 0.9980; sensitivity coefficient, 43 10^−3^ × min^−1^	[114], 2022
**Mediator-type amperometric BOD biosensors**			
Yeast *S. cerevisiae* immobilized in polyethylene terephthalate. Two-mediator system with ferricyanide and lipophilic mediator menadione	GGA/samples of river and sea water	Range of measured BOD_5_ values, 6.6–220 mg O_2_/dm^3^; *R* = 0.999; relative standard deviation, 1.3%; reduction of sensor response after the storage at 4 °C for 14 days, 93%; sensitivity, 0.33 μA/(mg O_2_/L)	[21], 2007
Bacteria *Exiguobacterium marius*, *B. horikoshii* and *Halomonas marina* immobilized in an organosilicon sol–gel matrix with PVA. Mediator of potassium hexacyanoferrate (III)	GGA/samples of sea water	Range of measured BOD_5_ values, 1.2–40 mg O_2_/dm^3^; BOD_5_ detection limit, 0.8 mg O_2_/dm^3^; standard deviation of the results, 3.8%; *R* = 0.988; sensitivity, 0.04 μA/(mg O_2_/L)	[16], 2008
Yeast *Candida krusei* applied to silica gel particles. Mediator of potassium hexacyanoferrate (III)	GGA, synthetic wastewaters (OECD), distillery wastewaters	Range of measured BOD5 values, 0–140 mg O_2_/dm^3^; measurement time, 20 min; sensitivity, 0.00625 mC/(mg BOD/L)	[115], 2010
Bacteria *Klebsiella pneumoniae*, mediator of potassium hexacyanoferrate (III)	GGA/synthetic wastewaters (OECD), municipal wastewaters	Linear dependence of BOD_5_ values, 30–500 mg O_2_/dm^3^ or 30–200 mg O_2_/dm^3^, using GGA and synthetic wastewaters, respectively; *R* = 0.9962 and 0.9934, respectively; sensitivity, 0.0016 μA/(mg BOD/L)	[47], 2011
*E. coli* bacteria immobilized in PVA–polyvinylpyridine copolymer. Mediator, neutral red	GGA/synthetic wastewaters (OECD), urea and real wastewaters	Range of measured BOD_5_ values, 50–1000 mg O_2_/dm^3^; relative standard deviation, 2.8%; measurement time, 15–60 min; sensitivity, 0.94 nA/(mg GGA/L)	[43], 2012
Activated sludge. Mediator, potassium hexacyanoferrate (III)	GGA/synthetic wastewaters (OECD), wastewaters of various origin	Range of measured BOD_5_ values, 2.1–40 mg O_2_/dm^3^; measurement time, 240 min; *R* = 0.96	[48], 2013
Biofilm based on wastewater microorganisms. Mediator, potassium hexacyanoferrate (III)	GGA/samples of treatment plant effluents	Range of measured BOD_5_ values, 2–200 mg O_2_/dm^3^; measurement time, 30 min; long-term stability, 60 days; sensitivity, 4.32 nA/mg BOD	[116], 2015
*Chromobacterium violaceum* bacteria immobilized in calcium alginate. Mediator, potassium hexacyanoferrate (III)	GGA/synthetic wastewaters (OECD), wastewaters from the food, textile and processing industries; wastewaters from landfills, river water	Range of measured BOD_5_ values, 20–230 mg O_2_/dm^3^; measurement time, 30 min; repeatability, 4.1%; *R* = 0.9663; sensitivity, 0.0538 nA/(mg O_2_/L)	[44], 2015
*P. aeruginosa* bacteria immobilized in polypyrrole. Mediator, potassium hexacyanoferrate (III)	GGA	Range of measured BOD_5_ values, 5–100 mg O_2_/dm^3^; BOD_5_ detection limit, 2 mg O_2_/dm^3^; repeatability, 3.6%; measurement time, 15 min; sensitivity, 1.58 nA/(mg O_2_/L)	[117], 2016
*P. aeruginosa* bacteria immobilized in the polypyrrole–alginate matrix. Mediator, semineutral red	Samples of river water	Range of measured BOD_5_ values, 5–100 mg O_2_/dm^3^; BOD_5_ detection limit, 3 mg O_2_/dm^3^; measurement time, 20 min; sensitivity, 3.67 nA/(mg O_2_/L)	[42], 2016
*B. subtilis* bacteria covalently bound to a gold electrode	GGA	Range of measured BOD_5_ values, 5–30 mg O_2_/dm^3^; BOD_5_ detection limit, 1.65 mg O_2_/dm^3^; *R* = 0.978; sensitivity, 1.58 nA/(mg O_2_/L)	[117], 2016
Yeast *D. hansenii* immobilized by adsorption on a graphite-paste electrode. Mediator, ferrocene	GGA/swamp water, wastewaters of municipal sewage treatment facilities, wastewaters of fermentation products	Range of measured BOD_5_ values, 25.2–320 mg O_2_/dm^3^; measurement time, 8–20 min; reproducibility, 2.2%; long-term stability, 39 days; *R* = 0.9971; sensitivity, 0.5 nA dm^3^ /mg	[46], 2017
*B. subtilis* bacteria in a three-dimensional porous graphene–polypyrrole (rGO-PPy) composite material. Mediator, potassium hexacyanoferrate (III)	Real samples of lake and river water, domestic wastewaters of Beijing	Range of measured BOD_5_ values, 4–60 mg O_2_/dm^3^; measurement time, 15 min; BOD_5_ detection limit, 1.8 mg O_2_/dm^3^; long-term stability, 60 days; sensitivity, 0.155 μA/(mg O_2_/L)	[45], 2017
Yeast *D. hansenii* immobilized by adsorption on a graphite-paste electrode. Two-mediator system, ferrocene–methylene blue	GGA, swamp and river water, wastewaters from municipal wastewater treatment plants	Range of measured BOD_5_ values, 2.5–7.2 mg O_2_/dm^3^; measurement time, 3.5–6 min; reproducibility, 1.2%; long-term stability, 43 days; *R* = 0.9913; sensitivity coefficient, 1.1 nA dm^3^/mg O_2_	[118], 2017
Activated sludge immobilized in a chitosan–bovine serum albumin (Chi–BSA) cryogel. Mediator, methylene blue in combination with graphene	GGA/wastewaters of various origins	Range of measured BOD_5_ values, 1–100 mg O_2_/dm^3^; BOD_5_ detection limit, 0.1 mg O_2_/dm^3^; operational stability, 2.9%; measurement time, 9 min; long-term stability, 65 days; sensitivity, 0.19 μA/(mg O_2_/L)	[51], 2017
*P. yeei* bacteria isolated from activated sludge immobilized by adsorption on a graphite-paste electrode. Mediator, ferrocene	GGA, river water	Range of measured BOD_5_ values, 1.3–200 mg O_2_/dm^3^; measurement time, 4–6 min; reproducibility, 2.9%; long-term stability, 22 days; *R* = 0.9934; sensitivity, 4.7 nA/(mg O_2_/dm^3^)	[119], 2019
Electroactive biofilm of *Sh. loihica.* Bidirectional extra-cellular electron transfer	Synthetic samples	Range of measured BOD_5_ values, 0–435 mg O_2_/dm^3^; assay time, less than 1 h; *R* = 0.99; sensitivity, 1.4 μA/(mg O_2_/dm^3^)	[28], 2020
Yeast *Blastobotrys adeninivorans* immobilized by adsorption on a graphite-paste electrode. Two-mediator system, ferrocene–neutral red	GGA/samples of surface waters	Range of measured BOD_5_ values, 0.16–2.7 mg O_2_/dm^3^; measurement time, 4–5 min; reproducibility, 1.5%; long-term stability, 26 days; *R* = 0.9693; sensitivity coefficient, 52 nA dm^3^/mg O_2_	[24], 2021
*P. yeei* bacteria isolated from activated sludge, immobilized in a redox-active polymer based on chitosan covalently bound to ferrocene and including carbon nanotubes	GGA/samples of surface waters	Range of measured BOD_5_ values, 0.1–4.5 mg O_2_/dm^3^; measurement time, 5 min; reproducibility, 1.8%; long-term stability, 50 days; *R* = 0.9916; sensitivity coefficient, 0.028 μA/(mg O_2_/dm^3^)	[57], 2021
Electroactive biofilms of activated sludge grown on the surface of a graphite-paste electrode modified with carbon nanotubes	GGA/samples of surface waters	Range of measured BOD_5_ values, 0.41–23 mg O_2_/dm^3^; measurement time, 5 min; reproducibility, 5.96%; long-term stability, 53 days; *R* = 0.9901; sensitivity coefficient, 0.04 μA/(mg O_2_/dm^3^)	[30], 2022
**BOD biosensors based on optical transducers**			
*B. subtilis* bacteria immobilized in a composite tetramethoxysilane sol–gel and chemically modified polyvinyl alcohol. Oxygen-sensitive film of tris(4,7-diphenyl-1,10-phenanthroline) ruthenium(II)	GGA/synthetic and household wastewaters	Range of measured BOD_5_ values, up to 25 mg O_2_/dm^3^; assay time, 15–30 min; long-term stability, over 45 days; *R* = 0.9808; sensitivity, 0.008 (mg/L min of dissolved oxygen)/(mg BOD/L)	[120], 2005
Microorganisms *B. licheniformis*, *Dietzia maris* and *Marinobacter marinus* from sea water, immobilized in polyvinyl alcohol and organosilicon sol–gel matrix. Sensitive film from organically modified silicate film with embedded oxygen-sensitive ruthenium complex	GGA/samples of sea water	Range of measured BOD_5_ values, from 0.2 up to 40 mg O_2_/dm^3^; measurement time, 3.2 min; stable operation time, up to 1 year; reproducibility, 2%; *R =* 0.9933	[15], 2006
*Photobacterium phosphoreum* bacteria immobilized in sodium alginate gel. Measurement of bioluminescence intensity	GGA/wastewaters of treatment plants	Assay time, 20 min; range of measured BOD_5_ values, 0 and 16 ppm; sensitivity, 522.75 conventional units/(mg BOD/L)	[65], 2007
Yeast *S. cerevisiae* immobilized by inclusion in PVA–styrilpyridinium gel. Optical sensor based on fluorescence quenching of a complex compound of ruthenium	GGA	Range of measured BOD_5_ values, 1–20 mg O_2_/dm^3^; *R* = 0.99; sensitivity, 0.001 conventional units/(mg GGA/L)	[66], 2015
Optical sensor based on tapered microfiber		Range of measured BOD_5_ values, 0.25–1 mg O_2_/dm^3^; BOD detection limit, 0.0016 mg O_2_/dm^3^	[121], 2020
**BOD biosensors based on microbial fuel cells**			
Activated sludge of sewage treatment plants	Synthetic wastewaters, real-time wastewater monitoring	Range of measured BOD_5_ values, up to 100 mg O_2_/dm^3^; assay time, ~60 min; reproducibility, 10% at the detection of BOD at a concentration of 100 mg O_2_/dm^3^	[122], 2005
Electrochemically active bacteria	Synthetic and real wastewaters	Linear dependence on BOD, up to 350 mg O_2_/dm^3^; stable operation time, 7 months; assay time, ~40 min; sensitivity coefficient, 0.001 mA/ppm COD	[123], 2009
Biofilm of wastewater microorganisms	Wastewaters	Range of measured BOD_5_ values, 17–78 mg O_2_/dm^3^; assay time, from 30 min up to 10 h; sensitivity, 3 (mA/m^2^)/(mg O_2_/L)	[124], 2011
Electroactive biofilm	Acetate, propionate, glucose, ethanol	Range of measured BOD_5_ values, 32–1280 mg O_2_/dm^3^; assay time, up to 20 h; sensitivity, 0.04 C/(mg O_2_/L)	[34], 2012
Electroactive biofilm of a mixture of six bacterial strains: *Thermincola carboxydiphila*, *P. aeruginosa*, *Ochrobactrum intermedium*, *Sh. frigidimarina*, *Citrobacter freundii*, *Clostridium acetobutylicum.*	Wastewaters	Range of measured BOD_5_ values, 8–240 mg O_2_/dm^3^; sensitivity, 0.7498 mV/(mg O_2_/L)	[18], 2014
Electroactive biofilm with a high content of *Geobacter* bacteria	Synthetic wastewaters	Range of measured BOD_5_ values, 174–1200 mg O_2_/dm^3^; assay time, 17.5 h; sensitivity, 0.0031 C/(mg O_2_/L)	[32], 2016
Electroactive biofilm of *Sh. loihica*	Synthetic wastewaters	Range of measured BOD_5_ values, 0–43.5 mg O_2_/dm^3^; sensitivity, 4.13 mV/(mg O_2_/L)	[125], 2018
*Gluconobacter oxydans* bacteria immobilized in N-vinylpyrrolidone-modified vinyl alcohol	GGA/wastewaters from food and biotechnological plants	Range of measured BOD_5_ values, 0.34–9.6 mg O_2_/dm^3^; measurement time, 30–130 min; reproducibility, 4.1%; long-term stability, 7 days; sensitivity coefficient, 8.3 mV dm^3^/mg	[80], 2018
Electroactive biofilm	Synthetic wastewaters	Range of measured BOD_5_ values, 25–500 mg O_2_/dm^3^; *R* = 0.995; sensitivity, 0.014 C/(mg O_2_/L)	[126], 2019
Electroactive biofilm	GGA/wastewaters with a high content of nitrates	Range of measured BOD_5_ values, 20–500 mg O_2_/dm^3^; sensitivity, 0.0355 C/(mg O_2_/L)	[127], 2020
Electroactive biofilm of *Sh. loihica*	GGA	Range of measured BOD_5_ values, 0–130.5 mg O_2_/dm^3^; *R* = 0.99; sensitivity, 2 μA/(mg O_2_/L)	[28], 2020
Electroactive biofilm	Synthetic wastewaters	Range of measured BOD_5_ values, up to 300 mg O_2_/dm^3^; long-term stability, over 30 days; *R* = 0.9950; sensitivity, 0.82 mA/(mg O_2_/L)	[84], 2020
Electroactive biofilm	Wastewaters from treatment facilities	Range of measured BOD_5_ values, 20–490 mg O_2_/dm^3^; assay time, 1.1 min; sensitivity, 2.9 μA/(g NaAc/L)	[128], 2020
Electroactive biofilm	GGA	Range of measured BOD_5_ values, 20–500 mg O_2_/dm^3^; long-term stability, about 30 days; *R* = 0.9950	[129], 2021
**BOD biosensors based on other methods of signal registration**			
Measurement of the concentration of CO_2_ produced by degradation by microorganisms of the carbon component of effluents. CO_2_ monitoring using an infrared spectrometer	Real-time measurements—determination of BOD in treatment facilities	*R* = 0.97457 (for CO_2_ and BOD concentrations); sensitivity, 12.99 ppm CO_2_/(mg BOD/L)	[87], 2005
Amperometric bioelectric “language” in a group cell using mathematical methods of data processing. Modification of electrodes by tyrosinase, horseradish peroxidase, acetylcholinesterase and butyrylcholinesterase	Wastewater samples	–	[88], 2005
Yeast *S. cerevisiae*. Photocolorimetric method in the presence of 2,6-dichlorophenolindophenol	GGA/samples of river water	Range of measured BOD_5_ values, 1.1–22 mg O_2_/dm^3^; reproducibility, 1.77%; long-term stability, 36 days; measurement time, 30 min; sensitivity, 0.0118 a.d./(mg O_2_/L)	[130], 2007
Activated sludge, pH converter. Determination of CO_2_ under aerobic conditions and NaOH under anaerobic conditions	Monitoring of the degree of contamination with organic compounds and toxicity	Sensitivity, 12.99 ppm CO_2_/(mg BOD/L)	[131], 2008
Magnetically functionalized *B. subtilis* bacteria on a matrix of ultramicroelectrodes. Amperometric determination of oxygen using palladium nanoparticles and reduced carboxyl graphene	Water samples	Range of measured BOD_5_ values, 2–15 mg O_2_/dm^3^; measurement time, 5 min; sensitivity, 2.093 nA/(mg O_2_/L)	[89], 2017
Yeast *S. cerevisiae*. Automated microfluidic chemiluminescent system with ferricyanide and quinone mediators	GGA/water samples	Range of measured BOD_5_ values, 10–315 mg O_2_/dm^3^	[132], 2018
A real-time respirometric system with photosensors and sensors for counting bubbles of released gas	GGA, municipal wastewaters	Range of measured BOD_5_ values, 0–420 mg O_2_/dm^3^; *R* = 0.99	[39], 2021

**Table 2 biosensors-12-00842-t002:** The main parameters of commercial BOD biosensors.

Model	Manufacturer, Country	Type	BOD Detection Range, mg/dm^3^	Assay Time, Min	Assay Error
Biox-1010	Endress and Hauser, Switzerland	Bioreactor	5–100,000	3–15	3%
BioMonitor	LAR, USA	Bioreactor	1–200,000	3–4	<5%
BOD-3000	Central Kagaku Corp., Japan	Bioreactor	0–1000	30–60	–
DKK BOD sensor 7842	DKK Corporation, Japan	Biosensor	0–60	5	10%
HABS-2000	KORBI, South Korea	Biofuel cell	0–200	10	5%
MB-DBO	Biosensores, Spain	Bioreactor	10–1000	30	<3%
Ra-BOD	AppliTek, Belgium	Bioreactor	20–100,000	30	<5%
QuickBOD α1000	Central Kagaku Corp., Japan	Biosensor	5–50	60	5%

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
