# Peer review of "Microbial Biosensors for Rapid Determination of Biochemical Oxygen Demand: Approaches, Tendencies and Development Prospects"

_biosensors, 2022, doi:10.3390/bios12100842_

Round 1
Reviewer 1 Report
Dear Authors:
This work proposes an extensive review on microbial biosensors for the determination of BOD, and the manuscript is also well written. As such, the matter is of interest, however, it suffers from two serious limits:
1) In chapter 3, some figures such as Figure 7 and Figure 15 are not clearly explained enough to describe the test principles or methods. Please revise carefully.
2)Please provide other reported reviews related to BOD biosensors, and compare this paper with those.
3) Please provide an overview figure.
Once the above concerns are fully addressed, the manuscript could be accepted for publication in this journal
Author Response
Response to Reviewer 1 Comments
This work proposes an extensive review on microbial biosensors for the determination of BOD, and the manuscript is also well written. As such, the matter is of interest, however, it suffers from two serious limits:
Point 1: In chapter 3, some figures such as Figure 7 and Figure 15 are not clearly explained enough to describe the test principles or methods. Please revise carefully.
Response 1: We added corresponding information in the text for better understanding of the presented figures:
Figure 7 (in the revised version, Fig. 6), lines 270-276: “Such a bioreactor has two holes for the influent and effluent and for the reference electrode. Also, there are the anode and cathode. The reactor is inoculated with sludge from a wastewater treatment plant to form a bioanode, and a nutrient medium for microbial growth is supplied. Application of a certain voltage across the electrodes enables tracing the changes in the electrochemical characteristics of the anode at the inflow of various carbon sources (including the investigated effluents), which serve as a substrate for microbial biochemical reactions.”
Figure 15 (in the revised version, Fig. 14), lines 616-652: “It is a membrane-free device in which two anodes (consisting of carbon fibre bundles forming a tree-like structure) with a reference electrode and a counter-electrode were placed in an aeration tank. Herewith, the diffuser for aeration of the medium was placed such that air bubbles from the diffuser prevented the adhesion of suspended solids from wastewater to the reference electrode, and provided their tight adherence to the anode. Due to intermittent aeration, an effective removal of pollutants from the wastewater was achieved. Bacteria present on the outside area of the covered anodes were considered to consume oxygen, by which anaerobic environments could be produced inside the covered anodes. These results demonstrated that the iBOB biosensor is capable of generating current from the wastewater in the aerating phase.”
Point 2: Please provide other reported reviews related to BOD biosensors, and compare this paper with those.
Response 2: The main feature of the presented review is that it considers biosensors for determining biochemical oxygen demand. Numerous reviews on microbial biosensors published over the recent years included chapters on BOD analysis; however, there have been no specialized works to cover modern approaches to BOD express analysis. The presented review is the first to consider all issues of BOD biosensor development: principles of signal generation, types of biomaterial, methods of its immobilization, approaches to sensor calibration, etc. Besides, the vast majority of reviews have been aimed at analysing scientific articles; in this review, we tried to collect and systematize information on commercial BOD analysers. It is also important to note that microbial biosensor reviews and older BOD biosensor reviews are discussed in the present review in sufficient detail (Refs. 7, 8, 12, 14, 29, 33, 35, 36, 82, 83, 91).
Point 3: Please provide an overview figure.
Response 3: We replaced two figures (Fig. 1 and Fig. 2) for an overview figure (Fig. 1A and B).

Reviewer 2 Report
The article can be accepted after minor revision
1. The authors need to explain the advantages of work over other work in the introduction part
2. Please discuss about receptor parameter
3. If possible please include sensitivity table
4. Please mention range of the concentration variation range
5. what are the specifications of the waste water
6. The re- 210 sponse time of such analyzers is usually less than 45 min, which makes it possible to mon- 211 itor wastewaters in real time—Please mention reference for above statement.
7. large area of the measuring electrode how you going justify
8. BOD electrochemical biosensors require anaerobic conditions for anode reac- 381 tions, they are not used directly in aerobic environments such as aeration tanks—more justification need
9. Immobilization techniques—possible include surfactant Immobilization techniques
10. Discuss about repetability of the biosensor
11. Commercially available BOD analyzers how differ more justification needed
12. Font size should be the same throughout the manuscript
Author Response
Response to Reviewer 2 Comments
The article can be accepted after minor revision.
Point 1. The authors need to explain the advantages of work over other work in the introduction part.
Response 1: The main feature of the presented review is that it considers biosensors for determining biochemical oxygen demand. Numerous reviews on microbial biosensors published over the recent years include chapters on BOD analysis; however, there have been no specialized works to cover modern approaches to BOD express analysis. The presented review is the first to consider all issues of BOD biosensor development: principles of signal generation, types of biomaterial, methods of its immobilization, approaches to sensor calibration, etc. Besides, the vast majority of reviews have been aimed at analysing scientific articles; in this review, we tried to collect and systematize information on commercial BOD analysers. It is also important to note that microbial biosensor reviews and older BOD biosensor reviews are discussed in the present review in sufficient detail (Refs. 7, 8, 12, 14, 29, 33, 35, 36, 82, 83, 91).
Point 2. Please discuss about receptor parameter.
Response 2:
We added the discussion of the characteristics after Table 1 (lines 000-000):
“Thus, to date, models of BOD biosensors based on a wide range of converter types are known. It can be noted that in recent years the number of works using methods of direct oxygen detection (Clark electrode or optical oxygen sensor) has significantly decreased. At the same time, a significant increase of works on the use of MFC for BOD analysis is observed, since such systems are characterized by a high correlation of the analysis results with the standard method and a very high stability. A common problem of many of the presented biosensor systems is a sufficiently high lower limit of the determined BOD5 values, which does not allow such systems to be used in the analysis of surface water samples in which BOD5 is usually within the range of 1 to 4 mg/dm3.”
Point 3. If possible please include sensitivity table.
Response 3: In Table 1, we added the values of biosensor sensitivity values in the cases when this is indicated in the text.
Point 4. Please mention range of the concentration variation range.
Response 4: In Table 1, we added the data on the BOD detection range by the biosensors in the cases when it is indicated in the cited publication.
Point 5. what are the specifications of the waste water.
Response 5: The specification of the wastewater depends on the type of enterprise and production. The legislation of each country establishes the wastewater parameters by special regulatory documents (some regulations are referred to in the review). Biosensor models for wastewater BOD determination are developed with respect to a specific production; often the receptor element uses the microorganisms contained in these wastewaters. For each receptor element, such a parameter as specificity is assessed, when the ability of microorganisms to oxidize various organic substrates is checked. A microorganism capable of oxidizing exactly the substrates that are contained in specific wastewater is chosen. In fact, the specificity of the microorganism is the main parameter for assessing the specification of wastewater.
Point 6. The response time of such analyzers is usually less than 45 min, which makes it possible to monitor wastewaters in real time—Please mention reference for above statement.
Response 6: For a biosensor, the time of a single response adds up mainly from the response development time and the time of receptor-element activity recovery (sensor washing). The duration of a single measurement depends on the nature of the biomaterial and the type of immobilization in the receptor element, as well as on the concentration of the substrate. The main data on the signal time are summed up in Table 1, in the biosensor parameters’ column (if they are given in the cited publication). From the data given it can be seen that the response time does not exceed 45 min.
Point 7. large area of the measuring electrode how you going justify.
Response 7: It is well known that the biosensor signal/current increases with the increase of the electrode surface area. To make our statement clearer, we changed the sentence “Due to the large area of the measuring electrode, significant currents, which may exceed the currents of the oxygen electrode, are generated in the mediator microbial sensors” to “Due to the large active surface area of the measuring electrode (which can be further increased by adding nanomaterials, etc.), on which more biological material can be immobilized, significant currents are generated in the mediator microbial sensors, that can exceed the currents of the oxygen electrode [51].” (lines 349-353).
Point 8. BOD electrochemical biosensors require anaerobic conditions for anode reac- 381 tions, they are not used directly in aerobic environments such as aeration tanks—more justification need.
Response 8: The following clarifications were added.
Lines 607-608:
Since BOD electrochemical biosensors based on Geobacter (an anaerobic bacterium widely used to develop biosensors and biofuel cells) require anaerobic conditions for anode reactions, they are not used directly in aerobic environments such as aeration tanks.
Line 613:
Geobacter spp. exoelectrogenic anaerobes were used to form a bioanode.
Point 9. Immobilization techniques—possible include surfactant Immobilization techniques.
Response 9: The determination of surfactants can lead to the leaching of biological material from the biosensor receptor. But when developing a biosensor, the substrate specificity of the biosensor is checked to determine which enzyme systems are present in the biological material, and such parameters as operational stability and long-term stability are also evaluated. Based on the results of this assessment, it is possible to make a judgement on the effect of individual reagents on the biosensor, whether a specific biosensor can be used to determine surfactants or they will destroy the receptor. As a rule, all these data, including data on the interfering factors, are present in the articles. Besides, with the cuvette method of measurement, as a rule, the sample is diluted, as a result of which the surfactant concentration in the cuvette can be 1:100, 1:1000, and its effect on the receptor will be reduced. We did not write about the influence of surfactants in our review, so as not to increase its volume.
Point 10. Discuss about repetability of the biosensor.
Response 10: The repeatability of the biosensor is related to the type of receptor element, the method of biological material immobilization, the type of wastewater used as a detected compound, as well as the presence of inhibitors. Certainly, as any other analytical method, biosensors have their limitations. At the same time, biosensors show fairly good data on the reproducibility and sustainability of the signal, which is shown in the articles given in our review as examples. Part of the data is given in the summarizing Table 1.
Point 11. Commercially available BOD analyzers how differ more justification needed.
Response 11: The differences in the presented commercial BOD analyzers are shown in sufficient detail in Table 2. However, we have somewhat expanded this section of the article and noted regulatory problems in the commercialization of BOD rapid analyzers.
After Figure 22, the following information was added: “An important issue of the active commercialization of BOD analyzers is the absence of or gaps in regulatory documentation in a number of countries. Certification of a new technique or a new device in different countries with different legislation can be a long and time-consuming process. Besides the delivery of the analyzers themselves, the manufacturer must establish a regular supply of biological material, which can also cause complications when crossing borders.”
Point 12. Font size should be the same throughout the manuscript.
Response 12: Font size was checked and corrected throughout the manuscript.

Reviewer 3 Report
Dear authors,
This is a well-written article, with a well-rounded literature review included. The authors have done a good job in including the industrial sensors/analyzers.
I have two suggestions/recommendations I have is towards the conclusion. Can you please on some of the challenges that need to be addressed about these sensing devices in the future.
The other suggestion I have is to include how in recent tech advancements AI has been integrated with such sensors which would be super helpful to the readers.
Author Response
Response to Reviewer 3 Comments
Point 1: I have two suggestions/recommendations I have is towards the conclusion. Can you please on some of the challenges that need to be addressed about these sensing devices in the future.
Response 1: This information was added before Conclusions (lines 1315-1320):
“An important issue of the active commercialization of BOD analyzers is the absence of or gaps in regulatory documentation in a number of countries. Certification of a new technique or a new device in different countries with different legislation can be a long and time-consuming process. Besides the delivery of the analyzers themselves, the manufacturer must establish a regular supply of biological material, which can also cause complications when crossing borders.”
Point 2: The other suggestion I have is to include how in recent tech advancements AI has been integrated with such sensors which would be super helpful to the readers.
Response 2: In Conclusions (lines 1340-1353), the following information was added: “Modern approaches to the development of biosensors and biofuel cells are associated with the active use of artificial intelligence (AI) [146,147]. These devices combine both wireless technologies and advanced machine learning algorithms, expanded diagnostic and data processing functions, ultimately leading to the decision-making level upon the analysis of the data obtained. In the future, such sensors can be used in remote and hard-to-reach areas to monitor water purity. Herewith, such devices will be able not only to signal about water quality, but also to start and stop additional wastewater treatment processes, as well as to make decisions about the possibility of these waters getting into the world ocean. In the long run, such systems based on MFC and AI can be used as alternative energy sources of autonomous action.”

Round 2
Reviewer 2 Report
Accept as it is